# EQUALIZED GENERATIVE TREATMENT: MATCHING $f$-DIVERGENCES FOR FAIRNESS IN GENERATIVE MODELS

## ABSTRACT

Fairness is a crucial concern for generative models, which not only reflect but can also amplify societal and cultural biases. Existing fairness notions for generative models are largely adapted from classification, focusing on balancing probability of generating each sensitive group. We show, both theoretically and empirically, that such criteria are brittle, as they can be satisfied even when different groups are modeled with widely varying quality. To address this gap, we introduce a new fairness definition for generative models, *equalized generative treatment* (EGT), which requires comparable generation quality across all sensitive groups, where this quality is measured via a reference $f$-divergence. We further analyze the trade-offs induced by EGT, showing that fairness constraints necessarily tie global model quality to the hardest group to approximate. Finally, we benchmark several strategies, including min-max optimization and group-conditional training, that directly target this criterion, and demonstrate through image generation experiments that EGT yields fairer outcomes without prohibitive losses in overall performance.

## 1 INTRODUCTION

The rise of generative models has revolutionized a wide range of domains, including natural language processing, computer vision, and scientific discovery Hu et al. (2025).As these systems become ever more ubiquitous, ensuring their *trustworthiness* has emerged as a critical challenge (Kucharavy et al., 2024). Trustworthiness is inherently multidimensional, encompassing robustness to perturbations (Carlini et al., 2024), protection of user privacy (Carlini et al.; 2023), and fairness (Choi et al., 2020; Zameshina et al., 2022). In fact, concerns about fairness are particularly pressing, as generative models not only reflect but also shape the ways information and cultural artifacts are created, distributed, and consumed. Left unchecked, these systems risk amplifying and existing societal biases and stereotypes. Despite growing interest, most approaches to fairness in generative models fail to provide definitions that reflect the unique characteristics of generative modeling. Existing metrics are often adapted directly from the literature on fair classification, focusing on recalibrating generative odds (i.e., the probability of being generated) across sensitive groups (Choi et al., 2020; Zameshina et al., 2022; Teo et al., 2022; 2023). However, such criteria fall short in capturing nuanced disparities on how different subpopulations are modeled, as illustrated in Figure 1 in the context of image generation. The proportion of images for each sensitive group (male/female) is identical, and the overall quality are similar. However, one model treats all sensitive groups equally, while the other produces lower quality and diversity results for the male class than for for the female class.

Beyond the evident failures discussed above, evaluating the quality of generative models remains an open problem. At its core, generative modeling aims to approximate a target data distribution with a model distribution. A common strategy is to minimize a statistical $f$-divergence, where the choice of $f$ determines the training objective and, consequently, the trained model (Goodfellow et al., 2014; Nowozin et al., 2016; Grover et al., 2018). After training, $f$-divergences are also used for evaluation, for example to compute the precision and recall of the model (Verine et al., 2023). While other evaluation metrics (e.g., FID (Heusel et al., 2017), MAUVE (Pillutla et al., 2023) or self-BLEU (Zhu et al., 2018)) can be used in practice, we adopt the general $f$-divergence framework both to remain consistent with existing training objectives and to provide a principled foundation for our fairness analysis, as further detailed in Section 2.1. In this context, we make the following contributions.

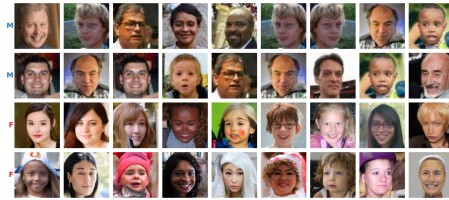

Figure 1: Illustrative example of two generative models with similar global evaluation metrics and balanced generative odds, but very different local metrics. In Set-Up 1 (left), images are slightly noised for both classes. In Set-Up 2 (right), the model generates noisy and redundant images for males while it generates high quality images and diverse samples for females. More details in Appendix B.1.

- **Contribution 1: Brittleness of existing fairness distribution.** In Section 3, we first show that existing notions of fairness can yield models that, while well-calibrated in terms of generative odds, can be arbitrarily unbalanced in terms of $f$-divergence between the target and trained conditional distribution for each sensitive group. This means that even when satisfying these definitions, models can generate some groups with much higher quality and diversity than some others. We validate this theoretical result with numerical experiments for image generation on the FFHQ dataset (Karras et al., 2020) or for text on the Wikipedia Biographies dataset (Bronnec et al., 2024). In each case, satisfying existing definitions do not prevent the generative model from treating each sensitive group differently in terms of generation quality.

- **Contribution 2: A new definition matching $f$-divergences.** Based on this observation, we present in Section 4 a new definition of fairness called *equalized generative treatment*, which relies on simultaneously controlling the difference in $f$-divergence between the target and trained conditional distribution for each sensitive group. We further show that applying this definition promotes the minimization of the highest $f$-divergence between conditionals among sensitive groups, which naturally leads us to study several optimization methods to improve fairness.

- **Contribution 3: Analyzing optimization methods for improved fairness.** As imbalances have been observed in real-world generative models, we propose to analyze the impact of several optimization strategies aimed at improving fairness with respect to EGT. We benchmark three approaches: loss reweighting introduced in (Choi et al., 2020), Min-Max optimization, and Conditional training. We first examine how these methods systematically balance the loss during training, and then assess their effect on EGT evaluation metrics, where we consistently observe improvements with at least one of the proposed methods.

## 2 BACKGROUND & RELATED WORKS

Let $\mathcal{X} \subset \mathbb{R}^d$, endowed with the euclidean norm $\|\cdot\|$. We denote $\mathcal{P}_\lambda(\mathcal{X})$ the set of probability distribution on $\mathcal{X}$ that are absolutely continuous w.r.t. the Lebesgue measure $\lambda$. For any $P \in \mathcal{P}_\lambda(\mathcal{X})$, its pdf (pdf) is denoted by $p = \frac{dP}{d\lambda}$. Finally, we denote $\Delta(\mathcal{A})$ the probability simplex over a finite set $\mathcal{A}$, and $\mathbb{1}_E(x)$ the indicator function of the event $x \in E$, for any $E \subset \mathcal{X}$.

### 2.1 $f$-DIVERGENCES IN GENERATIVE MODELING

Learning a generative model can be formalized as approximating a target distribution $P \in \mathcal{P}_\lambda(\mathcal{X})$ with a model distribution $Q$, where $Q$ belongs to an admissible family $\mathcal{Q} \subset \mathcal{P}_\lambda(\mathcal{X})$. To measure the discrepancy between $P$ and $Q$, we use the general class of $f$-divergences, defined below.

**Definition 2.1.** Let $f : (0, +\infty) \to (-\infty, +\infty]$ be a convex, lower semi-continuous function with $f(1) = 0$. For any $P, Q \in \mathcal{P}_\lambda(\mathcal{X})$, the $f$-divergence between $P$ and $Q$ is defined as

$$\mathcal{D}_f(P\|Q) = \int_{\text{SUPP}(Q)} f\left(\frac{p(x)}{q(x)}\right) q(x)\, d\lambda(x) + \bar{f}(\infty)\, P\left(\mathcal{X} \setminus \text{SUPP}(Q)\right),$$

where $p$ and $q$ are the pdfs of $P$ and $Q$, $\text{SUPP}(Q) = \{x \in \mathcal{X} \mid q(x) > 0\}$, $\bar{f}(\infty) = \lim_{t \to +\infty} f(t)/t$, and $f(0) = \lim_{t \to 0^+} f(t)$. This definition adopts the convention $0 \times \infty = 0$ to avoid the above being ill-defined when $\bar{f}(\infty) = \infty$ and $P\ (\mathcal{X} \setminus \text{SUPP}(Q)) = 0$.

The choice of $f$ in Definition 2.1 specifies the divergence we aim to minimize. For instance, the Kullback-Leibler divergence can be obtained by setting $f(t) = t \log t$. This divergence is usually minimized by likelihood-based methods used in LLMs Grattafiori et al. (2024), in Normalizing Flows (Rezende & Mohamed, 2016) or in some score-based diffusion models (Song et al., 2021). On the other hand, Generative Adversarial Networks (Goodfellow et al., 2014) typically optimize objective functions related to the Jensen-Shannon divergence, defined for $f(t) = t \log t - (t+1) \log(t+1)/2$. Additionally, several methods have also been proposed to minimize the Total Variation in GANs (Um & Suh, 2021) or LLMs (Ji et al., 2023). More generally, recent work has proposed modular frameworks that allow targeting a variety of $f$-divergences, enabling practitioners to tailor the objective to the specific context and application image modeling (Nowozin et al., 2016; Grover et al., 2018; Cai et al., 2020; Verine et al., 2023; Xu et al., 2025) and more recently for LLMs (Wang et al., 2023; Sun & Schaar, 2024; Go et al., 2023; Verine et al., 2025).

**Evaluation.** In practice, there exists many metrics to evaluate the performance of a generative model, each with their own strengths and weaknesses (Borji, 2022). Among these, the Fréchet Inception Distance (Heusel et al., 2017) or the Inception Score (Salimans et al., 2016) are widely used in image generation, while BLEU (Papineni et al., 2001) or ROUGE (Lin, 2004) are common in text generation. Among these metrics, a prominent example is the family of metrics that separately measure quality and diversity, most notably *precision* and *recall* for generative models Kynkäänniemi et al. (2019) refined by Kim et al. (2023). Given two distributions $P$ and $Q$, these metrics are defined as

$$\text{Precision}(Q\|P) = Q\left(\text{Supp}(P)\right), \qquad \text{Recall}(Q\|P) = P\left(\text{Supp}(Q)\right).$$

Precision and recall can be interpreted as a particular instance of $f$-divergences as demonstrated by Simon et al. (2019); Verine (2024). Furthermore, extensions and generalizations of these metrics building on $f$-divergences have been investigated in several recent works (Sajjadi et al., 2018; Djolonga et al., 2020; Pillutla et al., 2023).

## 2.2 Fairness in Generative Modeling

In the context of fairness, we extend this problem formulation by introducing a set of sensitive attributes $\mathcal{A}$ together with an oracle function $\psi : \mathcal{X} \to \mathcal{A}$ that maps each instance $x \in \mathcal{X}$ to its corresponding sensitive attribute. We assume $\psi$ is defined over the entire space, i.e., $\text{dom}(\psi) = \mathcal{X}$. This induces a partition of $\mathcal{X}$ into disjoint subsets $\mathcal{X}_a = \{x \in \mathcal{X} \mid \psi(x) = a\}$ for each $a \in \mathcal{A}$. For any distribution $P \in \mathcal{P}_\lambda(\mathcal{X})$ with pdf $p$, we can then express $P$ as the mixture

$$P = \sum_{a \in \mathcal{A}} \pi_a^P P_a \quad \text{with} \quad p_a(x) = \frac{p(x)}{\pi_a^P} \mathbb{1}_{\mathcal{X}_a}(x) \quad \forall x \in \mathcal{X},$$

where $\pi_a^P = P(\mathcal{X}_a)$ denotes the proportion of the population associated with attribute $a$, and $P_a$ is the conditional distribution restricted to $\mathcal{X}_a$. Thus, the vector $(\pi_a^P)_{a \in \mathcal{A}}$ lies in the simplex $\Delta(\mathcal{A})$, and $P$ can be interpreted as a mixture of attribute-conditional distributions. We call a distribution $P \in \mathcal{P}_\lambda(\mathcal{X})$ non-trivial if $\pi_a^P > 0$ for all $a \in \mathcal{A}$. In the following, we will always assume the target distribution we consider is non-trivial, since fairness considerations would otherwise be meaningless.

**Existing fairness criteria.** Most existing approaches to fairness in generative modeling focus on the proportions of sensitive attributes in the generated distribution. This perspective was popularized by Hutchinson & Mitchell (2019), who characterized fairness as the equal representation of a chosen sensitive attribute among generated samples. To the best of our knowledge, no standard terminology has been established in the literature for such proportion-based criteria. We therefore introduce the following nomenclature: *equalized generative odds* (EGO), and *matching generative odds* (MGO). The distinction between these two notions reflects different goals: EGO enforces uniform representation across groups, whereas MGO requires the generative model $Q$ to reproduce the proportions of the target distribution $P$.

**Definition 2.2.** Let $P, Q \in \mathcal{P}_\lambda(\mathcal{X})$ and $\delta > 0$. $Q$ is said to satisfy $\delta$-*equalized generative* odds if

$$\left|\pi_a^Q - \pi_{a'}^Q\right| \leq \delta, \quad \text{for all } a, a' \in \mathcal{A}.$$

When $\delta = 0$, we say that $Q$ satisfies *equalized generative odds*. Furthermore, $P$ and $Q$ are said to satisfy $\delta$-*matching generative odds* if

$$\left| \pi_a^Q - \pi_a^P \right| \le \delta, \quad \text{for all } a \in \mathcal{A}.$$

When $\delta = 0$, we say that $P$ and $Q$ satisfy *matching generative odds*, meaning that the group proportions under $Q$ exactly match those of $P$.

**Existing methods.** Most approaches to fairness in generative modeling have focused on enforcing group proportions, either through EGO or MGO. A variety of methods have been developed with this goal in mind. For instance, Choi et al. (2020) reweight the training distribution, enforcing EGO when the data is unbiased and approximating MGO otherwise. Other works modify the generation process itself, thereby directly controlling the proportion. For instance, Frankel & Vendrow (2018); Humayun et al. (2022); Tan et al. (2021) regularize the latent space to reduce deviations from MGO, while Zameshina et al. (2022) apply rejection sampling based on predicted sensitive attributes, which can be tuned to enforce either EGO or MGO. On the evaluation side, Teo et al. (2023) proposed classifier-based metrics designed to assess how well generative models satisfy EGO or MGO, but again the focus remained solely on proportions. Only very recently has work begun to go beyond this perspective. Mayer et al. (2024) introduced an evaluation based on the FID between majority and minority groups, although this was limited to the case of synthetic data generation. In summary, while most prior work has focused on improving or evaluating fairness through proportions (EGO or MGO), little attention has been given to local distributional discrepancies, which can be more effectively captured by $f$-divergences.

## 3  ON THE BRITTLENESS OF EXISTING DEFINITIONS

In Section 2.2, we discussed how most existing approaches to fairness in generative modeling are framed in terms of *matching generative odds* (MGO) or *equalized generative odds* (EGO). These notions capture fairness only at the level of group proportions, without accounting for the local behaviors (namely, how accurately the conditionals $(P_a)_{a \in \mathcal{A}}$ are reproduced). In this section, we argue that such a focus on proportions is inherently fragile. In particular, we show that even when MGO and EGO are perfectly satisfied, substantial discrepancies can remain in how different sensitive groups are modeled. We first illustrate this limitation through a simple example, then formalize it in a general theoretical result, and finally validate it empirically on state-of-the-art models.

### 3.1  THEORETICAL BRITTLENESS OF MATCHING AND EQUALIZED GENERATIVE ODDS

As a warm-up, consider the setting illustrated in Figure 2, where $\mathcal{X} = \mathbb{R}$, $\mathcal{A} = \{0, 1\}$, and the oracle function is $\psi = \mathbb{1}_{\mathbb{R}_+}$. The target distribution is defined as $P = \frac{1}{2}P_0 + \frac{1}{2}P_1$, where $P_0$ and $P_1$ are truncated Gaussian distributions with respective means $\pm 0.5$ and standard deviation $0.3$. As model family $\mathcal{Q}$, we consider the set of univariate Gaussian distributions $\{\mathcal{N}(\mu, \sigma^2) \mid (\mu, \sigma) \in \mathbb{R} \times \mathbb{R}_+\}$ that we rescale on each side of the origin to enforce MGO and EGO. To examine the potential disparities arising under these criteria, we focus on the Jensen Shannon divergence $\mathcal{D}_{\mathrm{JS}}$. For this divergence, we study the level set $\{Q \in \mathcal{Q} \mid \mathcal{D}_{\mathrm{JS}}(P\|Q) = \epsilon\}$, for a fixed level $\epsilon = 1$.

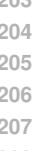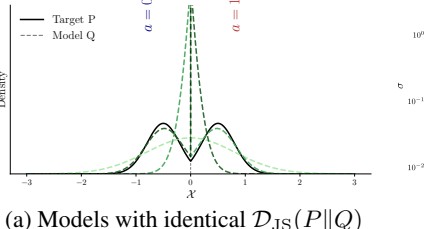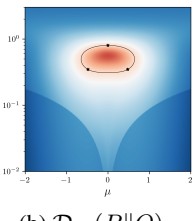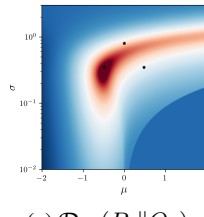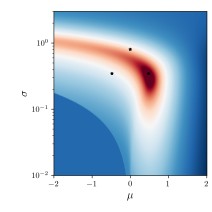

(a) Models with identical $\mathcal{D}_{\mathrm{JS}}(P\|Q)$    (b) $\mathcal{D}_{\mathrm{JS}}(P\|Q)$    (c) $\mathcal{D}_{\mathrm{JS}}(P_0\|Q_0)$    (d) $\mathcal{D}_{\mathrm{JS}}(P_1\|Q_1)$

Figure 2: Jensen–Shannon divergence between a target distribution $P$ and rescaled Gaussian models $Q = \mathcal{N}(\mu, \sigma^2)$. (2a) Models with the same global divergence $\mathcal{D}_{\mathrm{JS}}(P\|Q)$ can still differ greatly. (2b) Level set for $\mathcal{D}_{\mathrm{JS}} = 1$, with selected models marked by stars. (2c)–(2d) Conditional divergences for the two groups, models on the same level set may yield highly unbalanced conditional divergences.

Figure 2b shows that this level set allows for several pairs of admissible parameters $(\mu, \sigma) \in \mathbb{R} \times \mathbb{R}_+$, with three representative solutions highlighted by points marked with a star. On the one hand, for the upper star in Figure 2b, the conditional divergences $\mathcal{D}_{\mathrm{JS}}(P_0 \| Q_0)$ and $\mathcal{D}_{\mathrm{JS}}(P_1 \| Q_1)$ are both small and almost identical. On the other hand, at the left and right stars in Figure 2b, a clear imbalance appears. Indeed, in both cases, one of the conditional divergences is as low as $0.02$, while the other reaches $1.98$. This illustrative example shows that even when MGO and EGO are perfectly satisfied, the model can still exhibit arbitrarily poor fidelity for one of the sensitive groups.

**More general result.** The brittleness observed above is not a mere artifact of our toy example, but rather a general phenomenon. Even when $\mathcal{Q}$ is allowed to range over all distributions in $\mathcal{P}_\lambda(\mathcal{X})$ that satisfy EGO and MGO with $P$, and even when the global $f$-divergence $\mathcal{D}_f(P \| Q)$ is constrained to be arbitrarily small (but non-zero), it remains possible to construct situations in which one group incurs an arbitrarily larger conditional divergence than the others. To make this precise, we introduce the *$f$-divergence level set* around $P$ as

$$\mathcal{S}_{\mathcal{D}_f}(P, \epsilon) \;=\; \big\{ Q \in \mathcal{P}_\lambda(\mathcal{X}) \;\big|\; \mathcal{D}_f(P \| Q) = \epsilon \big\}.$$

By working within $\mathcal{P}_\lambda(\mathcal{X})$, we impose no restriction on model expressivity, thereby going beyond the specificity of our toy example. In this setting, the phenomenon of imbalanced conditional divergences across attributes can be formalized as follows.

**Theorem 3.1.** *Let $P \in \mathcal{P}_\lambda(\mathcal{X})$ be a non-trivial target distribution satisfying EGO, and let $f$ be a continuous function such that $\mathcal{D}_f$ defines an $f$-divergence. For any $\epsilon \in (0, f(0) + \bar{f}(+\infty))$ and any $\gamma \in (0, \epsilon)$, there exists $Q^\gamma \in \mathcal{S}_{\mathcal{D}_f}(P, \epsilon)$ such that $Q^\gamma$ satisfies MGO with $P$, but for which there exists $\bar{a} \in \mathcal{A}$ with*

$$\mathcal{D}_f(P_{\bar{a}} \| Q_{\bar{a}}^\gamma) \;\geq\; \mathcal{D}_f(P_a \| Q_a^\gamma) + \gamma, \qquad \text{for all } a \in \mathcal{A} \setminus \{\bar{a}\}.$$

In other words, even when both EGO and MGO are met, it is always possible to build a model performing arbitrarily worse on one group than on the others. This establishes that global fairness criteria based solely on proportions provide no guarantee of conditional quality for the groups.

### 3.2 Observing the brittleness in practice

The brittleness highlighted above is not only theoretical, it also manifests in practice when working with off-the-shelf generative models. To demonstrate this, we conduct a set of experiments using state-of-the-art pretrained models, as well as two representative fairness-enhancing methods from the literature: (i) the reweighting (RW) method of Choi et al. (2020), which modifies the loss function to encourage EGO, and (ii) the rejection sampling (RS) method of Zameshina et al. (2022), which can enforce either EGO or MGO by filtering generated samples. We evaluate these approaches across both image and text generation tasks. In the main text, we focus on reporting fairness-relevant evaluation metrics, specifically disparities in precision and recall. For completeness, additional results together with a detailed description of the experimental setup, the evaluation process, and additional results for diffusion models and LLMs are provided in Appendix B.3 and Appendix B.4.

**Diffusion models.** We first consider diffusion models, focusing on the EDM-VP architectures introduced by Karras et al. (2022), and trained on the FFHQ dataset Karras et al. (2019). The sensitive attribute is gender (male/female), with proportions approximately $44/56$ in the training set. We evaluate the pretrained model, as well as variants fine-tuned with RW, and models generating via RS to enforce MGO or EGO. On Figure 3, we report precision and recall for the different models, along with their corresponding $\delta$-MGO and $\delta$-EGO values (in %) in Table 3c. We observe that pretrained models are naturally close to MGO ($\delta$-MGO = 0.005) and that rejection sampling effectively enforces both EGO and MGO. However, despite these proportion-based improvements, we consistently find discrepancies in precision and recall between groups up to 2.2% in precision and 4.5% in recall, demonstrating that fairness with respect to conditional divergences remain unaddressed.

**Large Language Models.** We next consider the chat (instruction tuned) version of LLaMA-3.2 model (Grattafiori et al., 2024) finetuned on Wikipedia Biographies. We consider both the 1B and 3B parameter variants. The dataset is derived from the Wikipedia Biographies dataset introduced by Bronnec et al. (2024), which contains header of biographies of individuals. Examples are given in

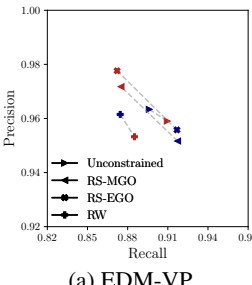

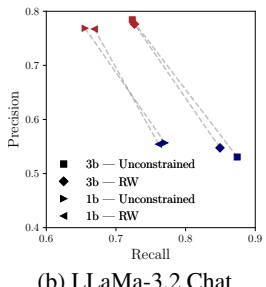

| Models | Method | $\delta$-MGO | $\delta$-EGO |
|---|---|---|---|
| EDM-VP | Pretrained | 0.5 | 5.8 |
| | RW | 1.4 | 3.9 |
| | RS-MGO | 0.2 | - |
| | RS-EGO | - | 0.1 |
| llama-3.2-1b Chat | Pretrained | 6.9 | 31.4 |
| | RW | 7.1 | 21.6 |
| llama-3.2-3b Chat | Pretrained | 8.2 | 32.7 |
| | RW | 9.0 | 33.5 |

| (a) EDM-VP | (b) LLaMa-3.2 Chat | (c) $\delta$-MGO & $\delta$-EGO (in %) |
|---|---|---|

Figure 3: Precision and recall for EDM-VP on FFHQ (3a) and for llama-3.2 Chat 1b and 3b on the Wikipedia Biographies (3b) under three settings: pretrained, with reweighting (RW) to promote EGO, and with rejection sampling (RS) to enforce MGO or EGO. The table (3c) reports the corresponding $\delta$-MGO and $\delta$-EGO values (in %) for all models considered in the experiments. We observe that while RW and RS effectively improve EGO and MGO, respectively, significant discrepancies in precision and recall persist across groups, demonstrating the brittleness of proportion-based definitions.

the Appendix B.4.3. The sensitive attribute is also gender, with proportions approximately 75/15. While the pretrained models are not particularly close to MGO, over-representing male biographies, we observe that precision and recall discrepancies largely exceeds the ones observed for diffusion models, reaching up to 25.37% in precision and 15.22% in recall, as shown in Figure 3b.

# 4    EQUALIZED GENERATIVE TREATMENT (EGT)

The analysis in Section 3 shows that fairness criteria based solely on group proportions, such as MGO and EGO, are inherently brittle. To address this limitation, we introduce a new criterion, called *equalized generative treatment*, that explicitly requires parity across sensitive groups in terms of $f$-divergences. We then show that applying this definition promotes the minimization of the highest $f$-divergence between conditionals among sensitive groups.

## 4.1    DEFINITION AND FIRST PROPERTY

We now introduce the notion of *equalized generative treatment* (EGT). This criterion provides a stronger, more fine-grained notion of fairness by explicitly linking generative quality to each subgroup using a reference $f$-divergence. Formally, EGT can be stated as follows.

**Definition 4.1.** Let $P, Q \in \mathcal{P}_\lambda(\mathcal{X})$ and let $f$ be such that $\mathcal{D}_f$ defines an $f$-divergence. For any $\delta > 0$, we say that $Q$ and $P$ satisfy $\delta$-*equalized generative treatment* w.r.t. $\mathcal{D}_f$ if

$$\left| \mathcal{D}_f(P_a \| Q_a) - \mathcal{D}_f(P_{a'} \| Q_{a'}) \right| \leq \delta, \qquad \text{for all } a, a' \in \mathcal{A}.$$

When $\delta = 0$, we say that $Q$ and $P$ satisfy *equalized generative treatment* w.r.t. $\mathcal{D}_f$.

**Conditional closure.** Conceptually, the best way to approximate a target distribution $P$ while respecting $\delta$-EGT would be to treat each sensitive group independently. More specifically, for every $a \in \mathcal{A}$, one would learn a conditional distribution $Q_a$ that minimizes $\mathcal{D}_f(P_a \| Q_a)$, and then recombine the conditional models using the true proportions $(\pi_a^P)_{a \in \mathcal{A}}$ of the target distribution. This procedure would ensure that each sensitive group is treated equally and that the final mixture aligns with $P$ as closely as possible. In practice, however, generative models are rarely designed in such a conditional manner. Instead, they typically produce distributions $Q$ as a whole, without the ability to freely optimize and reassemble conditionals. As a result, the typical family $\mathcal{Q}$ of candidate models used in practice seldom captures what would be achievable if sensitive group-level flexibility were available. To reason about this gap, we introduce the *conditional closure* of $\mathcal{Q}$, an augmented set of distributions that allows explicit recombination of conditionals.

**Definition 4.2.** Let $\mathcal{Q} \subseteq \mathcal{P}_\lambda(\mathcal{X})$ be a family of candidate models. For each $a \in \mathcal{A}$, let $\mathcal{Q}_a := \{R \in \mathcal{P}_\lambda(\mathcal{X}_a) \mid \exists Q \in \mathcal{Q} \text{ with } Q_a = R\}$ be the set of conditional distributions for group $a$ within $\mathcal{Q}$. Let also $\Delta_\mathcal{Q} := \{(\pi_a)_{a \in \mathcal{A}} \in \Delta(\mathcal{A}) \mid \exists Q \in \mathcal{Q} \text{ with } (\pi_a^Q)_{a \in \mathcal{A}} = (\pi_a)_{a \in \mathcal{A}}\}$ be the set of group

proportions in $\mathcal{Q}$. Then the *conditional closure* of $\mathcal{Q}$, denoted $\overline{\mathcal{Q}}_{\mathcal{A}}$, is defined as the set of distributions whose conditionals belong to the sets $(\mathcal{Q}_a)_{a \in \mathcal{A}}$ and proportions lie in $\Delta_{\mathcal{Q}}$, i.e.,

$$\overline{\mathcal{Q}}_{\mathcal{A}} := \left\{ Q \in \mathcal{P}_\lambda(\mathcal{X}) \;\middle|\; Q = \sum_{a \in \mathcal{A}} \pi_a^Q Q_a, \text{ with } Q_a \in \mathcal{Q}_a \; \forall a \in \mathcal{A}, \text{ and } (\pi_a^Q)_{a \in \mathcal{A}} \in \Delta_{\mathcal{Q}} \right\}.$$

Intuitively, $\overline{\mathcal{Q}}_{\mathcal{A}}$ can be seen as the completed version of $\mathcal{Q}$ for the fairness-constrained problem. It extends $\mathcal{Q}$ by allowing to select conditional models for each sensitive group from $\mathcal{Q}$ and then reassemble them under any group proportions realizable by $\mathcal{Q}$. In this sense, $\overline{\mathcal{Q}}_{\mathcal{A}}$ represents a best-case modeling scenario. If the original set $\mathcal{Q}$ has been explicitly designed to support attribute-conditional subdivision, then we may have $\mathcal{Q} = \overline{\mathcal{Q}}_{\mathcal{A}}$. In general, however, $\overline{\mathcal{Q}}_{\mathcal{A}}$ strictly contains $\mathcal{Q}$, since most generative models are not conditionally structured for sensitive attributes. The role of $\overline{\mathcal{Q}}_{\mathcal{A}}$ is to provide a principled baseline for understanding the limits of generative modeling under fairness constraints. Theorem 4.3 shows that, under the $\delta$-EGT, the global divergence $\mathcal{D}_f(P|Q)$ is bounded below by the largest conditional divergence across sensitive groups in is the best model within $\overline{\mathcal{Q}}_{\mathcal{A}}$.

**Theorem 4.3.** *Let $P \in \mathcal{P}_\lambda(\mathcal{X})$ be a non-trivial target probability distribution and $f$ be a function such that $\mathcal{D}_f$ defines an $f$-divergence. Let $\mathcal{Q}$ be set of candidate probability distributions satisfying MGO with $P$ and such that there exists $Q^\star \in \arg\min_{Q \in \overline{\mathcal{Q}}_{\mathcal{A}}} \mathcal{D}_f(P\|Q)$. Then for any $\delta > 0$, if $Q \in \mathcal{Q}$ and $P$ satisfy $\delta$-EGT w.r.t. $\mathcal{D}_f$, then*

$$\mathcal{D}_f(P\|Q) \geq \max_{a \in \mathcal{A}} \mathcal{D}_f(P_a \| Q_a^\star) - \delta.$$

This result highlights the fact that enforcing EGT promotes the minimization of the highest $f$-divergence between conditionals among groups, which naturally leads us to the following.

## 4.2 THEORETICALLY ENFORCING EGT

Before diving into empirical considerations in the next section, we first analyze the feasibility of training a model that satisfies EGT from a theoretical viewpoint. To do so, we consider a simple theoretical setting where $\mathcal{X} = \mathbb{R}$, $\mathcal{A} = \{0, 1\}$, $\psi = \mathbb{1}_{\mathbb{R}_+}$ and $P$ is designed as a truncated mixture of 4 Gaussian with respective means $-4, -2.7, 2, 4$ and common standard deviation $0.3$. Furthermore, we set the class of candidate model to be $\mathcal{Q} := \left\{ \frac{1}{2}\mathcal{N}(\mu_1, \sigma_1^2) + \frac{1}{2}\mathcal{N}(\mu_2, \sigma_2^2) \mid (\mu_1, \mu_2, \sigma_1, \sigma_2) \in \mathbb{R}^2 \times \mathbb{R}_+^2 \right\}$, rescaled as in Section 3. The parameters $(\mu_1, \mu_2, \sigma_1, \sigma_2)$ are optimized to approximate $P$, but while regularizing the objective function to enforce $\delta$-EGT. Specifically, we minimize the objective

$$\min_{Q \in \mathcal{Q}} \mathcal{D}_f(P\|Q) + \lambda \sum_{a, a' \in \mathcal{A}} \left| \mathcal{D}_f(P_a\|Q_a) - \mathcal{D}_f(P_{a'}\|Q_{a'}) \right|,$$

where $\lambda \geq 0$ controls the strength of the EGT regularization. $\lambda = 0$ recovers standard divergence minimization, while larger values place increasing emphasis on balancing the conditional divergences across groups. Using `scipy.optimize.minimize`, we solve this problem for $\lambda \in \{0, 1, 100\}$ and report the results in Figure 4. Increasing $\lambda$ balances the conditional divergences but only by raising them to the level of the worst-performing group. This highlights the fundamental tension of EGT: enforcing conditional parity inevitably trades off overall performance.

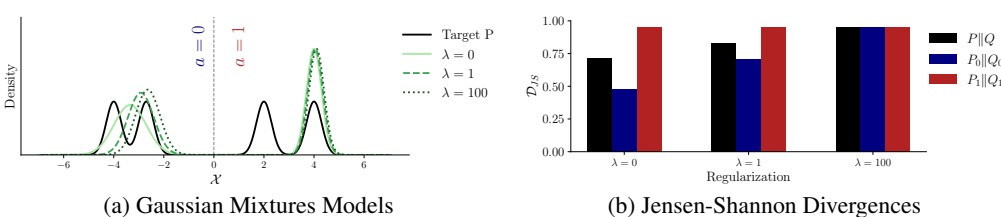

(a) Gaussian Mixtures Models      (b) Jensen-Shannon Divergences

Figure 4: Effect of EGT regularization in the Gaussian mixture setting. (4a) Target distribution $P$ (black) and optimized models $Q$ for different values of $\lambda \in \{0, 1, 100\}$. (4b) Jensen–Shannon divergences for the full distribution and for each group. Larger $\lambda$ reduces disparity between groups but drives all divergences toward the worst case, illustrating the trade-off inherent in enforcing EGT.

While conceptually appealing, directly enforcing EGT via $f$-divergence regularization is impractical in modern training loops. $f$-divergences are inherently one-sided, yielding either a tractable minimization objective (variational) or a maximization surrogate (adversarial), but not both simultaneously (Huszár, 2015; Arjovsky & Bottou, 2017). These limitations motivate the next section, where we study practical approximations to EGT rather than relying on direct regularization.

## 5 Improved fairness through EGT

In this section, we investigate three optimization strategies aimed at improving fairness with respect to EGT. These methods are architecture-agnostic and applicable to both image and text generative models. While direct EGT regularization is theoretically well-founded, it is computationally impractical; the approaches below provide feasible alternatives. Specifically, we consider: 1) loss reweighting, 2) Min–Max training, and 3) conditional training, which are formalized as follows.

$$1)\ \min_{Q\in\mathcal{Q}} \mathcal{D}_f\Big(\sum_{a\in\mathcal{A}} \tfrac{1}{|\mathcal{A}|} P_a\|Q\Big) \qquad 2)\ \min_{Q\in\mathcal{Q}}\max_{a\in\mathcal{A}} \mathcal{D}_f\left(P_a\|Q_a\right) \qquad 3)\ \min_{\substack{(Q_a)_{a\in\mathcal{A}}\in\prod_a \mathcal{P}_\lambda(\mathcal{X}_a)\\ \text{st.}Q=\sum_a \pi_a^P Q_a\in\mathcal{Q}}} \mathcal{D}_f(P\|Q).$$

Loss reweighting was introduced by Choi et al. (2020) and rescales the training loss according to group proportions, effectively reducing the dominance of majority groups. Min–Max training instead minimizes the worst conditional divergence across groups, pushing the model toward balancing its errors. Finally, conditional training equips the model with explicit group conditioning. Note that, while the idealized formulation allows each $Q_a$ to be learned independently, this is prohibitively expensive in practice. To circumvent this, we adopt an alternative where groups share most parameters through a conditional neural network (see Appendices B.3.2 and B.4.1 for details).

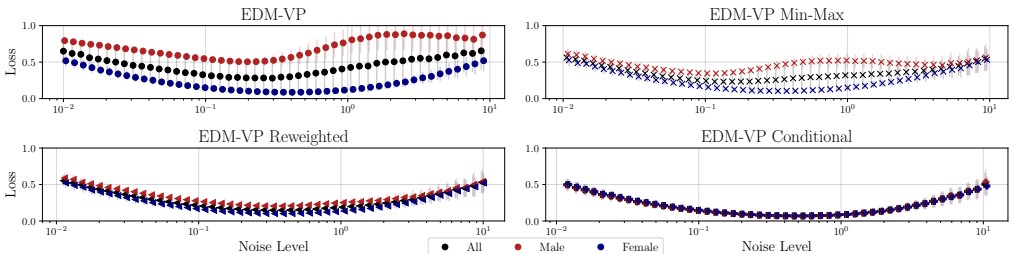

Figure 5: Estimated denoising losses per noise level for EDM-VP trained on FFHQ. Baseline exhibits a persistent gap between male and female groups. Reweighting and Min–Max reduce the gap, while conditional training almost eliminates it.

**Effect on the training loss.** We first analyze the effect of these methods on the diffusion training objective. For EDM-VP on FFHQ, the loss is a weighted sum of denoising score matching losses at multiple noise levels. Figure 5 shows the estimated loss at each level for the baseline, reweighted, Min–Max, and conditional training. All three methods narrow the gap between male and female groups, with conditional training nearly closing it entirely.

**Observing EGT evaluation metrics.** The complete set of evaluation metrics for both diffusion models and LLMs is provided in Appendix B.3.3 and Appendix B.4.2. We summarize the key findings in Table 1 for diffusion and Table 2 for LLMs, where we report $\delta$-EGT in terms of precision, recall, and their sum (a valid $f$-divergence by linearity), together with $\delta$-FID for completeness. For the unconditional EDM-VP model, both reweighting and Min–Max substantially reduce disparities across settings (unconstrained, MGO, and EGO). Min–Max achieves the strongest overall gains: for example, under EGO rejection sampling, $\delta$-Precision and $\delta$-Recall drop from 2.2 and 4.5 in the baseline to 0.0 and 0.3, yielding a $\delta$-PR of only 0.4. Reweighting is also effective, particularly under EGO where it reduces $\delta$-Recall to 0.2. Both methods consistently lower $\delta$-FID, with values as low as 0.02–0.06 compared to 0.21–0.37 for the pretrained model. While conditional variants achieve very competitive $\delta$-FID scores (down to 0.00–0.17), their $\delta$-PR values remain significantly higher (typically 6–9) than those observed in the unconditional setting. This indicates that, conditional training does not automatically improve fairness gaps in terms of EGT. For LLMs, both reweighting and Min–Max also improve fairness over pretrained baselines. On the 1B model, Min–Max reduces

Table 1: Evaluation of the different optimization methods on the FFHQ dataset with EDM-VP model. We report the difference in precision, recall and the sum between male and female. All results are in percentage points. The best results are in bold per method generation method and per metric.

| | | (a) Unconditional | | | | (b) Conditional | | | |
| Train | Generation | $\delta$-Precision | $\delta$-Recall | $\delta$-PR | $\delta$-FID | $\delta$-Precision | $\delta$-Recall | $\delta$-PR | $\delta$-FID |
|---|---|---|---|---|---|---|---|---|---|
| Pretrained | Unconst. | 0.4 | 1.3 | 1.8 | 0.37 | - | - | - | - |
| | MGO | 2.0 | 4.2 | 6.2 | 0.34 | 1.6 | 6.8 | 8.4 | **0.08** |
| | EGO | 2.2 | 4.5 | 6.7 | 0.21 | 1.7 | 7.7 | 9.5 | **0.00** |
| RW | Unconst. | 0.8 | 1.1 | 1.9 | **0.07** | - | - | - | - |
| | MGO | 0.6 | **1.3** | **1.9** | 0.13 | **1.2** | 5.9 | **7.2** | 0.26 |
| | EGO | 0.3 | **0.2** | 0.5 | **0.02** | 0.9 | **5.0** | **6.0** | 0.17 |
| Min-Max | Unconst. | **0.3** | **0.2** | **0.5** | 0.28 | - | - | - | - |
| | MGO | **0.4** | 1.6 | 2.0 | **0.06** | 1.5 | **5.7** | **7.2** | 0.48 |
| | EGO | **0.0** | 0.3 | **0.4** | 0.06 | **0.8** | 6.6 | 7.3 | 0.38 |

$\delta$-Precision from 21.18 to 19.55 and $\delta$-Recall from 11.66 to 9.57, yielding a $\delta$-PR of 29.12 (down from 32.85). On the 3B model, reweighting achieves the best recall reduction (15.22 → 12.34), while Min–Max slightly outperforms on precision (25.37 → 22.27), leading to balanced improvements in $\delta$-PR (40.59 → 35.21). In the conditional setting, improvements are even more consistent: for example reducing $\delta$-PR from 37.19 to 32.03. These results highlight that, unlike in the diffusion setting, conditional training reliably enhances fairness.

*Disclaimer.* These fairness improvements do not come for free. As discussed in Section 4, reducing disparities typically occurs at the expense of the best-performing group. This trade-off is visible in the attribute-conditional results reported in Appendix B, where improvements in fairness coincide with performance deterioration for the strongest subgroup.

# 6 CONCLUDING REMARKS

In this work, we demonstrated that existing proportion-based criteria for fairness in generative modeling are inherently brittle: even when perfectly satisfied, a model's output quality can remain arbitrarily unbalanced across sensitive groups. To address this, we introduced equalized generative treatment (EGT), a fairness definition that enforces comparable $f$-divergences across groups, and studied its applicability. While $f$-divergences are among the most widely used metrics in generative models, alternatives such as the FID or integral probability metrics also exist and merit further investigation. In particular, it would be interesting to examine whether our analysis (especially results analogous to Theorems A.3 and A.4) remains valid in these alternative settings. Similarly, we expect that our results could extend to training schemes that directly minimize Wasserstein distances, such as Wasserstein GANs. We leave these explorations to future work.

Table 2: Evaluation of the different optimization methods used to finetune LLaMa3.2 Chat 1b and 3b on the Wikipedia Biographies dataset. We report the difference in precision/recall between subgroups. All results are in percentage points. The best results per model are in bold.

| | | (a) Unconditional | | | | (b) Conditional | | |
| Model | Training | $\delta$-MGO | $\delta$-Precision | $\delta$-Recall | $\delta$-PR | $\delta$-Precision | $\delta$-Recall | $\delta$-PR |
|---|---|---|---|---|---|---|---|---|
| 1b | Pretrained | **6.9** | 21.18 | 11.66 | 32.85 | **19.09** | **9.19** | **28.27** |
| | RW | 7.1 | 21.29 | 10.06 | 31.34 | 20.70 | 11.24 | 31.95 |
| | Min-Max | 7.0 | **19.55** | **9.57** | **29.12** | 20.36 | 9.20 | 29.56 |
| 3b | Pretrained | **8.2** | 25.37 | 15.22 | 40.59 | 24.31 | 12.88 | 37.19 |
| | RW | 9.0 | 22.88 | **12.34** | **35.21** | **21.94** | **10.09** | **32.03** |
| | Min-Max | 9.2 | **22.27** | 13.24 | 35.50 | 25.96 | 12.67 | 38.62 |

## LLM USAGE

LLMs were used solely to polish writing and improve the fluency of text drafted by the authors. No scientific ideas, analyses, or results were generated by LLMs. Additionally, code development benefited from standard code-completion tools such as GitHub Copilot. The authors take full responsibility for all content.

## REPRODUCIBILITY STATEMENT

All details necessary to reproduce the Gaussian setups are provided in the main text. The full procedures for large-scale experiments are described in Appendix B. Source code to reproduce our experiments, including fine-tuning of diffusion models, building the Wikipedia Biographies Dataset, LoRA fine-tuning of LLMs, will be released at the time of publication. Currently, the code is available in the OpenReview Supplementary Materials.

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

# A   MATHEMATICAL SUPPLEMENTARY MATERIAL

## A.1   USEFUL LEMMA ON THE SURJECTIVITY OF $\mathcal{D}_f$

We begin with a statement that is of independent interest, as it establishes the surjectivity of the mapping $\mathcal{D}_f(R, \cdot)$ under suitable continuity assumptions on $f$. This result is stated in Lemma A.1.

**Lemma A.1.** *Let $f : [0, +\infty) \to (-\infty, +\infty]$ be a continuous function such that $\mathcal{D}_f$ defines an $f$-divergence. Then for any $R \in \mathcal{P}_\lambda(\mathcal{X})$, the map $\mathcal{D}_f(R, \cdot) : \{Q \in \mathcal{P}_\lambda(\mathcal{X}) \mid \text{SUPP}(R) = \text{SUPP}(Q)\} \to \mathbb{R}_+$ is surjective onto the interval $(0, f(0) + \bar{f}(+\infty))$.*

*Proof.* Let $R \in \mathcal{P}_\lambda(\mathcal{X})$, and let $f : [0, +\infty) \to (-\infty, +\infty]$ be a convex and continuous function with $f(1) = 0$. Let us also fix $\alpha \in (0, 1)$ and $\beta \in (1, +\infty)$. Since $R$ is absolutely continuous with respect to the Lebesgue measure $\lambda$, its cumulative density function is also continuous on $\mathbb{R}$. Accordingly, by definition of the cumulative density function and by the intermediate value theorem, there exists $A_\alpha \in \mathcal{B}(\mathcal{X})$ such that $R(A_\alpha) = \alpha$.

*1) Construction of an auxiliary mapping $\phi$.* Let us denote by $r := \frac{dR}{d\lambda}$ the probability density function of $R$ with respect to $\lambda$. Thanks to the above, we can define $Q_\alpha^\beta$ the probability distribution in $\mathcal{P}_\lambda(\mathcal{X})$ that admits a probability density function $\frac{dQ_\alpha^\beta}{d\lambda} = q_\alpha^\beta$ defined for all $x \in \mathcal{X}$ as

$$q_\alpha^\beta(x) := \frac{1}{\beta}\left(\frac{1}{\alpha}r(x)\mathbb{1}_{A_\alpha}(x)\right) + \left(1 - \frac{1}{\beta}\right)\left(\frac{1}{1-\alpha}\,r(x)\mathbb{1}_{\mathcal{X}\setminus A_\alpha}(x)\right). \tag{1}$$

By construction, since $\beta > 1$ and $\alpha \in (0, 1)$, $q_\alpha^\beta$ is a valid probability density function and $Q_\alpha^\beta \in \mathcal{P}_\lambda(\mathcal{X})$. Furthermore, also by construction we have $\text{SUPP}(R) = \text{SUPP}(Q_\alpha^\beta)$ (see Appendix A.1.1 for more details). Hence, using the alternative characterization of $f$-divergences in the special case of matching support (see e.g. (Polyanskiy & Wu, 2025, Chapter 7)), we have

$$\mathcal{D}_f(R\|Q_\alpha^\beta) = \int_{\mathcal{X}} f\left(\frac{r(x)}{q_\alpha^\beta(x)}\right) q_\alpha^\beta(x)\,d\lambda(x) \quad \text{with the convention } f(\tfrac{0}{0}) \times 0 = 0.$$

Using the above, and by definition of $Q_\alpha^\beta$, we thus have

$$\mathcal{D}_f(R\|Q_\alpha^\beta) = \int_{A_\alpha} f\left(\frac{r(x)}{\frac{r(x)}{\beta\alpha}}\right)\frac{r(x)}{\alpha\beta}\,d\lambda(x) + \int_{\mathcal{X}\setminus A_\alpha} f\left(\frac{r(x)}{r(x)\frac{\beta-1}{(1-\alpha)\beta}}\right)\frac{\beta-1}{\beta(1-\alpha)}r(x)\,d\lambda(x)$$

$$= \int_{A_\alpha} \frac{f(\beta\alpha)}{\alpha\beta}r(x)\,d\lambda(x) + \int_{\mathcal{X}\setminus A_\alpha} f\left(\frac{(1-\alpha)\beta}{\beta-1}\right)\frac{\beta-1}{\beta(1-\alpha)}r(x)\,d\lambda(x).$$

Furthermore, by linearity of the integral and by construction of $A_\alpha$, we have

$$\mathcal{D}_f(R\|Q_\alpha^\beta) = \frac{1}{\alpha\beta}f(\beta\alpha)\int_{A_\alpha} r(x)\,d\lambda(x) + \frac{\beta-1}{\beta(1-\alpha)}f\left(\frac{(1-\alpha)\beta}{\beta-1}\right)\int_{\mathcal{X}\setminus A_\alpha} r(x)\,d\lambda(x)$$

$$= \frac{1}{\alpha\beta}f(\beta\alpha)\,R(A_\alpha) + \frac{\beta-1}{\beta(1-\alpha)}f\left(\frac{(1-\alpha)\beta}{\beta-1}\right)R(\mathcal{X}\setminus A_\alpha)$$

$$= \frac{1}{\beta}f(\beta\alpha) + \frac{\beta-1}{\beta}f\left(\frac{(1-\alpha)\beta}{\beta-1}\right). \tag{2}$$

Since the $\alpha$, and $\beta$ have been chosen arbitrarily. The above construction holds for any $(\alpha, \beta) \in (0, 1) \times (1, +\infty)$. Thus, we can define $\phi : (0, 1) \times (1, +\infty) \to \mathbb{R}_+$ the mapping such that

$$\phi(\alpha, \beta) := \mathcal{D}_f(R\|Q_\alpha^\beta) = \frac{1}{\beta}f(\beta\alpha) + \frac{\beta-1}{\beta}f\left(\frac{(1-\alpha)\beta}{\beta-1}\right), \; \forall(\alpha, \beta) \in (0, 1) \times (1, +\infty).$$

As $\{Q_\alpha^\beta \mid (\alpha, \beta) \in (0, 1) \times (1, +\infty)\} \subseteq \{Q \in \mathcal{P}_\lambda(\mathcal{X}) \mid \text{SUPP}(R) = \text{SUPP}(Q)\}$, to get the expected result it suffices to show that $\phi$ is surjective onto the interval $(0, f(0) + \bar{f}(+\infty))$.

*2) Studying the surjectivity of $\phi$.* As $f$ is a continuous function, by composition of continuous functions, $\phi$ is jointly continuous on $(0,1) \times (1,+\infty)$. Accordingly, still using the intermediate value theorem, $\phi$ is surjective onto $[\phi(\alpha_a, \beta_a), \phi(\alpha_b, \beta_b)]$ for any $\alpha_a, \alpha_b \in (0,1)$ and $\beta_a, \beta_b \in (1,+\infty)$. In particular, since this holds for any choice of $\alpha_a, \alpha_b, \beta_a$, and $\beta_b$ we also have that $\phi$ is surjective onto $[\phi(1/2, 2), \lim_{\substack{\alpha \to 1 \\ \beta \to +\infty}} \phi(\alpha, \beta))$. To conclude, we just need to compute each of these terms:

- $\phi(1/2, 2) = \frac{1}{2}f(1) + \frac{1}{2}f(1) = 0$, and

- $\lim_{\substack{\alpha \to 1 \\ \beta \to +\infty}} \phi(\alpha, \beta) = \lim_{\beta \to \infty} \frac{1}{\beta}f(\beta) + \lim_{\beta \to +\infty} \frac{\beta-1}{\beta}f(0) = \bar{f}(+\infty) + f(0).$

*3) Conclusion.* By surjectivity of $\phi$ and by construction of $\{Q_\alpha^\beta \mid (\alpha, \beta) \in (0,1) \times (1,+\infty)\}$, we just showed that for any $y \in (0, f(0) + \bar{f}(\infty))$, there exists $Q \in \{Q \in \mathcal{P}_\lambda(\mathcal{X}) \mid \mathrm{SUPP}(R) = \mathrm{SUPP}(Q)\}$ such that $\mathcal{D}_f(R, Q) = y$, which concludes the proof. $\square$

### A.1.1 ADDITIONAL SANITY CHECKS FOR THE CONSTRUCTION OF $Q_\alpha^\beta$ IN LEMMA A.1

**Well-definiteness of the distribution.** Let us first show that for any $(\alpha, \beta) \in (0,1) \times (1,+\infty)$, $q_\alpha^\beta$ is a valid probability density function. For this, first note that the terms $1/\beta$, $1 - 1/\beta$, $1/\alpha$ and $1/(1-\alpha)$ are all positive. Furthermore, $r$ is itself a probability density function by definition, hence non-negative. Hence, by construction $q_\alpha^\beta$ is a non-negative function. Also note that, for any $(\alpha, \beta) \in (0,1) \times (1,+\infty)$, integrating against $\lambda$ over $\mathcal{X}$, we get

$$\int_{\mathcal{X}} q_\alpha^\beta(x)\, d\lambda(x) = \int_{\mathcal{X}} \frac{1}{\beta}\left(\frac{1}{\alpha}r(x)\mathbb{1}_{A_\alpha}(x)\right) + \left(1 - \frac{1}{\beta}\right)\left(\frac{1}{1-\alpha}\, r(x)\mathbb{1}_{\mathcal{X}\backslash A_\alpha}(x)\right) d\lambda(x)$$

$$= \int_{A_\alpha} \frac{1}{\beta\alpha}r(x)d\lambda(x) + \int_{\mathcal{X}\backslash A_\alpha} \left(1 - \frac{1}{\beta}\right)\frac{1}{1-\alpha}\, r(x)d\lambda(x)$$

Which by linearity of the integral and definition of $r$ and $A_\alpha$ gives

$$= \frac{1}{\beta\alpha}R(A_\alpha) + \left(1 - \frac{1}{\beta}\right)\frac{1}{1-\alpha}R(\mathcal{X}\backslash A_\alpha) = \frac{1}{\beta} + \left(1 - \frac{1}{\beta}\right) = 1.$$

**Matching supports.** Let us now show that $\mathrm{SUPP}(Q_\alpha^\beta) = \mathrm{SUPP}(R)$. To do so let us first consider $x \notin \mathrm{SUPP}(R)$, by definition of $q_\alpha^\beta$ we have

$$q_\alpha^\beta(x) = \frac{1}{\beta\alpha}r(x)\mathbb{1}_{A_\alpha}(x) + \left(1 - \frac{1}{\beta}\right)\frac{1}{1-\alpha}\, r(x)\mathbb{1}_{\mathcal{X}\backslash A_\alpha}(x).$$

Since $x \notin \mathrm{SUPP}(R)$, we have $r(x) = 0$, hence $q_\alpha^\beta(x) = 0$. Accordingly, $x \notin \mathrm{SUPP}(Q_\alpha^\beta)$. This means by contrapositive that $\mathrm{SUPP}(Q_\alpha^\beta) \subset \mathrm{SUPP}(R)$. Similarly, let $x \notin \mathrm{SUPP}(Q_\alpha^\beta)$ we have

$$q_\alpha^\beta(x) = \frac{1}{\beta\alpha}r(x)\mathbb{1}_{A_\alpha}(x) + \left(1 - \frac{1}{\beta}\right)\frac{1}{1-\alpha}\, r(x)\mathbb{1}_{\mathcal{X}\backslash A_\alpha}(x) = 0.$$

Since all terms $1/\beta$, $1 - 1/\beta$, $1/\alpha$ and $1/(1-\alpha)$ are positive, this means that $r(x) = 0$. Hence $x \notin \mathrm{SUPP}(R)$, which gives us $\mathrm{SUPP}(R) \subset \mathrm{SUPP}(Q_\alpha^\beta)$.

### A.2 PROOF OF THEOREM 1

Before proceeding to the proof of Theorem 1, we state a central lemma that decomposes the $f$-divergence between two measures $P$ and $Q$ in terms of a linear combination of the $f$-divergence between their marginals. The result is given in Lemma A.2

### A.2.1 PRELIMINARY LEMMA

**Lemma A.2.** *Let $P \in \mathcal{P}_\lambda(\mathcal{X})$ be a non-trivial target probability distribution, $Q \in \mathcal{P}_\lambda(\mathcal{X})$, and $f$ be function such that $\mathcal{D}_f$ defines an $f$-divergence. If $P$ and $Q$ have matching generative odds, then*

$$\mathcal{D}_f(P\|Q) = \sum_{a \in \mathcal{A}} \pi_a^Q \mathcal{D}_f(P_a\|Q_a),$$

*where the decomposition according to $\mathcal{A}$ is as defined in Section 2.2.*

*Proof.* Let $P \in \mathcal{P}_\lambda(\mathcal{X})$ be a target probability distribution and $f$ be a function such that $\mathcal{D}_f$ defines an $f$-divergence. Let $Q \in \mathcal{P}_\lambda(\mathcal{X})$, such that $P$ and $Q$ have matching generative odds. By definition of the attribute mapping $\psi$, $\{\mathcal{X}_a \mid a \in \mathcal{A}\}$ is a partition of $\mathcal{X}$. Hence, denoting $p = \frac{dP}{d\lambda}$ and $q = \frac{dQ}{d\lambda}$ the respective probability density functions of $P$ and $Q$, we have

$$\mathcal{D}_f(P\|Q) = \int_{\text{SUPP}(Q)} f\left(\frac{p(x)}{q(x)}\right) q(x) d\lambda(x) + \bar{f}(+\infty) P\left(\mathcal{X} \setminus \text{SUPP}(Q)\right)$$

$$= \sum_{a \in \mathcal{A}} \int_{\mathcal{X}_a \cap \text{SUPP}(Q)} f\left(\frac{p(x)}{q(x)}\right) q(x) d\lambda(x) + \bar{f}(+\infty) P\left(\mathcal{X} \setminus \text{SUPP}(Q)\right).$$

Note that, for any $a \in \mathcal{A}$, by definition we have

$$\mathcal{X}_a \cap \text{SUPP}(Q) := \{x \in \mathcal{X}_a \mid q(x) > 0\} = \{x \in \mathcal{X} \mid q(x) \times \mathbb{1}_{\mathcal{X}_a}(x) > 0\}.$$

Now recall that we assumed $\pi_a^P > 0$ for any $a \in \mathcal{A}$. Furthermore, $Q$ is assumed to match the odds of $P$, meaning that $\pi_a^Q = \pi_a^P > 0$ for all $a \in \mathcal{A}$. Hence we have

$$\mathcal{X}_a \cap \text{SUPP}(Q) = \{x \in \mathcal{X} \mid q(x) \times \mathbb{1}_{\mathcal{X}_a}(x) > 0\}$$

$$= \{x \in \mathcal{X} \mid \tfrac{q(x)}{\pi_a^Q} \times \mathbb{1}_{\mathcal{X}_a}(x) := q_a(x) > 0\} = \text{SUPP}(Q_a). \tag{3}$$

Using (3) in the first decomposition of $\mathcal{D}_f$, we get

$$\mathcal{D}_f(P\|Q) = \sum_{a \in \mathcal{A}} \int_{\text{SUPP}(Q_a)} f\left(\frac{p(x)}{q(x)}\right) q(x) d\lambda(x) + \bar{f}(+\infty) P\left(\mathcal{X} \setminus \text{SUPP}(Q)\right).$$

Now recall that we can always rewrite $p$ and $q$ as mixtures of attribute-conditional probability density functions $p = \sum_{a \in \mathcal{A}} \pi_a^P p_a$ and $q = \sum_{a \in \mathcal{A}} \pi_a^Q q_a$ where for any $a \in \mathcal{A}$, $q_a(x) = p_a(x) = 0$ for all $x \notin \mathcal{X}_a$. Accordingly, the $f$-divergence between $P$ and $Q$ can be rewritten as

$$\mathcal{D}_f(P\|Q) = \sum_{a \in \mathcal{A}} \int_{\text{SUPP}(Q_a)} f\left(\frac{\pi_a^P p_a(x)}{\pi_a^Q q_a(x)}\right) \pi_a^Q q_a(x) d\lambda(x) + \bar{f}(+\infty) P\left(\mathcal{X} \setminus \text{SUPP}(Q)\right). \tag{4}$$

Note also that, by definition of the by definition of the attribute oracle, we have $P = \sum_{a \in \mathcal{A}} \pi_a^P P_a$ where $\text{SUPP}(P_a) \subset \mathcal{X}_a$. Using this we can rewrite $P\left(\mathcal{X} \setminus \text{SUPP}(Q)\right)$ as

$$P\left(\mathcal{X} \setminus \text{SUPP}(Q)\right) = \sum_{a \in \mathcal{A}} \pi_a^P P_a(\mathcal{X} \setminus \text{SUPP}(Q)) = \sum_{a \in \mathcal{A}} \pi_a^P P_a(\mathcal{X}_a \setminus \text{SUPP}(Q))$$

$$= \sum_{a \in \mathcal{A}} \pi_a^P P_a(\mathcal{X}_a \setminus (\text{SUPP}(Q) \cap \mathcal{X}_a))$$

$$= \sum_{a \in \mathcal{A}} \pi_a^P P_a(\mathcal{X}_a \setminus \text{SUPP}(Q_a)).$$

Where the last line comes from using (3). Using the above in (4) we get

$$\mathcal{D}_f(P\|Q) = \sum_{a \in \mathcal{A}} \int_{\text{SUPP}(Q_a)} f\left(\frac{\pi_a^P p_a(x)}{\pi_a^Q q_a(x)}\right) \pi_a^Q q_a(x) d\lambda(x) + \bar{f}(+\infty) \sum_{a \in \mathcal{A}} \pi_a^P P_a\left(\mathcal{X}_a \setminus \text{SUPP}(Q_a)\right).$$

Finally, Using the matching odds property ,i.e., the fact that $\pi_a^P = \pi_a^Q$ for all $a \in \mathcal{A}$, we have

$$
\mathcal{D}_f(P\|Q) = \sum_{a \in \mathcal{A}} \int_{\text{SUPP}(Q_a)} f\left(\frac{p_a(x)}{q_a(x)}\right) \pi_a^Q q_a(x) d\lambda(x) + \bar{f}(+\infty) \sum_{a \in \mathcal{A}} \pi_a^Q P_a\left(\mathcal{X}_a \setminus \text{SUPP}(Q_a)\right)
$$

$$
= \sum_{a \in \mathcal{A}} \pi_a^Q \left(\int_{\text{SUPP}(Q_a)} f\left(\frac{p_a(x)}{q_a(x)}\right) q_a(x) d\lambda(x) + \bar{f}(+\infty) P_a\left(\mathcal{X}_a \setminus \text{SUPP}(Q_a)\right)\right)
$$

$$
= \sum_{a \in \mathcal{A}} \pi_a^Q \mathcal{D}_f(P_a\|Q_a), \text{ which concludes the proof.}
$$

$\square$

### A.2.2 Proof of the Theorem

We now turn to the proof of Theorem 1, restated below as Theorem A.3. The argument relies primarily on Lemma A.1 and Lemma A.2.

**Theorem A.3.** *Let $P \in \mathcal{P}_\lambda(\mathcal{X})$ be a target probability distribution satisfying equalized generative odds and $f$ be a continuous function such that $\mathcal{D}_f$ defines an $f$-divergence. For any $\epsilon \in (0, f(0) + \bar{f}(+\infty))$ and $\gamma \in (0, \epsilon)$, there exists $Q^\gamma \in \mathcal{S}_{\mathcal{D}_f}(P, \epsilon)$ satisfying matching generative odds with $P$, but for which there exists $\bar{a} \in \mathcal{A}$ such that*

$$
\mathcal{D}_f(P_{\bar{a}}\|Q_{\bar{a}}^\gamma) \geq \mathcal{D}_f(P_a\|Q_a^\gamma) + \gamma, \text{ for all } a \in \mathcal{A} \setminus \{\bar{a}\}.
$$

*Proof.* Let $P \in \mathcal{P}_\lambda(\mathcal{X})$ be a target probability distribution satisfying equalized generative odds and $f$ be a continuous function such that $\mathcal{D}_f$ defines an $f$-divergence. By definition of the attribute mapping $\psi$, $\{\mathcal{X}_a \mid a \in \mathcal{A}\}$ is a partition of $\mathcal{X}$. Hence we can rewrite $P = \sum_{a \in \mathcal{A}} \pi_a^P P_a$ with $\text{SUPP}(P_a) \subset \mathcal{X}_a$. Let us now fix $\epsilon \in (0, f(0) + \bar{f}(+\infty))$, $\gamma \in (0, \epsilon)$ and let $\bar{a} \in \mathcal{A}$. Since $f$ is continuous, by Lemma A.1, we know there exists $Q_{\bar{a}}^\gamma \in \mathcal{P}_\lambda(\mathcal{X})$ such that $\text{SUPP}(P_{\bar{a}}) = \text{SUPP}(Q_{\bar{a}}^\gamma)$ and $\mathcal{D}_f(P_{\bar{a}}\|Q_{\bar{a}}^\gamma) = \epsilon + \gamma \frac{|\mathcal{A}|-1}{|\mathcal{A}|}$. Similarly, for any $a \in \mathcal{A} \setminus \{\bar{a}\}$ there exist a distribution $Q_a^\gamma \in \mathcal{P}_\lambda(\mathcal{X})$ such that $\text{SUPP}(P_a) = \text{SUPP}(Q_a^\gamma)$ and $\mathcal{D}_f(P_a\|Q_a^\gamma) = \epsilon - \frac{\gamma}{|\mathcal{A}|}$. Using these, we define $Q^\gamma$ as

$$
Q^\gamma = \frac{1}{|\mathcal{A}|} \sum_{a \in \mathcal{A}} Q_a^\gamma.
$$

Let us now consider $Q^\gamma$ as defined above. Recall that the target distribution $P$ is assumed to satisfy equalized odds, hence $\pi_a^P = \frac{1}{|\mathcal{A}|}$ for all $a \in \mathcal{A}$. Furthermore, by definition, $\pi_a^{Q^\gamma} = \frac{1}{|\mathcal{A}|} = \pi_a^P$ for all $a \in \mathcal{A}$, which means that $Q^\gamma$ satisfies matching odds with $P$. Hence, using Lemma A.2, we have

$$
\mathcal{D}_f(P\|Q^\gamma) = \sum_{a \in \mathcal{A}} \pi^{Q_a^\gamma} \mathcal{D}_f(P_a\|Q_a^\gamma) = \frac{1}{|\mathcal{A}|} \sum_{a \in \mathcal{A}} \mathcal{D}_f(P_a\|Q_a^\gamma) = \epsilon
$$

Nevertheless, by construction, we also have

$$
\mathcal{D}_f(P_{\bar{a}}\|Q_{\bar{a}}) \geq \mathcal{D}_f(P_a\|Q_a) + \gamma.
$$

Hence, by construction, we just designed a probability distribution $Q^\gamma$ satisfying the conditions of Theorem A.3. $\square$

### A.3 Proof of Theorem 2

We now present the proof of Theorem 2, restated below as Theorem A.4. The proof relies primarily on Lemma A.2 and the notion of conditional closure.

**Theorem A.4.** *Let $P \in \mathcal{P}_\lambda(\mathcal{X})$ be a target probability distribution and $f$ be a function such that $\mathcal{D}_f$ defines an $f$-divergence. Let $\mathcal{Q}$ be set of candidate probability distributions satisfying matching odds with respect to $P$ and such that there exists $Q^\star \in \arg\min_{Q \in \overline{\mathcal{Q}}_\mathcal{A}} \mathcal{D}_f(P\|Q)$. Then for any $\delta > 0$, if $Q \in \mathcal{Q}$ satisfies $\delta$-equalized generative treatment of $P$ with respect to $\mathcal{D}_f$, then*

$$
\mathcal{D}_f(P\|Q) \geq \max_{a \in \mathcal{A}} \mathcal{D}_f(P_a\|Q_a^\star) - \delta.
$$

*Proof.* Let $P \in \mathcal{P}_\lambda(\mathcal{X})$ be a target probability distribution satisfying equalized generative odds and $f$ be a function such that $\mathcal{D}_f$ defines an $f$-divergence. By definition of the attribute mapping $\psi$, $\{\mathcal{X}_a \mid a \in \mathcal{A}\}$ is a partition of $\mathcal{X}$. Hence we can rewrite $P = \sum_{a \in \mathcal{A}} \pi_a^P P_a$ with $\text{SUPP}(P_a) \subset \mathcal{X}_a$. Let us now consider $Q \in \mathcal{Q}$. As all probability distributions in $\mathcal{Q}$ satisfy matching odds with respect to $P$, we use Lemma A.2 to write

$$\mathcal{D}_f(P\|Q) = \sum_{a \in \mathcal{A}} \pi_a^Q \mathcal{D}_f(P_a, Q_a). \tag{5}$$

*Reasoning ab absurdum.* Based on the above, we reason ab absurdum to obtain the expected result. Let us take $a_\# \in \arg\max_{a \in \mathcal{A}} \mathcal{D}_f(P_a\|Q_a^\star)$, where $Q^\star \in \arg\min_{Q \in \overline{\mathcal{Q}}_\mathcal{A}} \mathcal{D}_f(P\|Q)$, and assume that

$$\mathcal{D}_f(P\|Q) < \mathcal{D}_f\left(P_{a_\#}\|Q_{a_\#}^\star\right) - \delta. \tag{6}$$

Note that, by definition of $\delta$-equalized generative treatment, we know that

$$\mathcal{D}_f\left(P_{a_\#}\|Q_{a_\#}\right) - \mathcal{D}_f\left(P_a\|Q_a\right) \leq \delta, \ \forall a \in \mathcal{A}.$$

Hence, we also have $\mathcal{D}_f\left(P_{a_\#}\|Q_{a_\#}\right) \leq \mathcal{D}_f\left(P_a\|Q_a\right) + \delta, \ \forall a \in \mathcal{A}$. Multiplying both sides by $\pi_a^Q$ for all $a \in \mathcal{A}$ and summing the terms one gets

$$\sum_{a \in \mathcal{A}} \pi_a^Q \mathcal{D}_f\left(P_{a_\#}\|Q_{a_\#}\right) \leq \sum_{a \in \mathcal{A}} \pi_a^Q \left(\mathcal{D}_f\left(P_a\|Q_a\right) + \delta\right).$$

Finally, using the fact that $\sum_{a \in \mathcal{A}} \pi_a^Q = 1$ and the decomposition in (5), we obtain

$$\mathcal{D}_f\left(P_{a_\#}\|Q_{a_\#}\right) \leq \mathcal{D}_f\left(P\|Q\right) + \delta. \tag{7}$$

Now, let us consider the distribution $\tilde{Q} \in \mathcal{P}_\lambda(\mathcal{X})$ defined as

$$\tilde{Q} = \sum_{\substack{a \in \mathcal{A} \\ a \neq a_\#}} \pi_a^{Q^\star} Q_a^\star + \pi_{a_\#}^{Q^\star} Q_{a_\#}.$$

By definition of $\overline{\mathcal{Q}}^\mathcal{A}$, we have that $\tilde{Q} \in \overline{\mathcal{Q}}^\mathcal{A}$ and that $Q^\star$ satisfies matching generative odds with respect to $P$. Hence, $\tilde{Q}$ also satisfies matching generative odds with respect to $P$, which gives us (using Lemma A.2)

$$\mathcal{D}_f\left(P\|\tilde{Q}\right) = \sum_{\substack{a \in \mathcal{A} \\ a \neq a_\#}} \pi_a^{Q^\star} \mathcal{D}_f(P_a\|Q_a^\star) + \pi_{a_\#}^{Q^\star} \mathcal{D}_f(P_{a_\#}\|Q_{a_\#}).$$

Substituting successively (7) and (6) in the above, and using the fact that $Q^\star$ satisfies matching generative odds with respect to $P$ (to obtain the last equality), we have

$$\mathcal{D}_f\left(P\|\tilde{Q}\right) \leq \sum_{\substack{a \in \mathcal{A} \\ a \neq a_\#}} \pi_a^{Q^\star} \mathcal{D}_f(P_a\|Q_a^\star) + \pi_{a_\#}^{Q^\star} \left(\mathcal{D}_f(P\|Q) + \delta\right)$$

$$< \sum_{\substack{a \in \mathcal{A} \\ a \neq a_\#}} \pi_a^{Q^\star} \mathcal{D}_f(P_a\|Q_a^\star) + \pi_{a_\#}^{Q^\star} \mathcal{D}_f\left(P_{a_\#}\|Q_{a_\#}^\star\right)$$

$$< \sum_{a \in \mathcal{A}} \pi_a^{Q^\star} \mathcal{D}_f(P_a\|Q_a^\star) = \mathcal{D}_f\left(P\|Q^\star\right).$$

According to the above, we just constructed a probability distribution $\tilde{Q} \in \overline{\mathcal{Q}}^\mathcal{A}$ that has an $f$-divergence strictly smaller than $Q^\star$ with respect to $P$. By definition of $Q^\star$, this is impossible, hence contradicting our initial assumption that $\mathcal{D}_f(P\|Q) < \mathcal{D}_f\left(P_{a_\#}\|Q_{a_\#}^\star\right) - \delta$. In particular, this means that

$$\mathcal{D}_f(P\|Q) \geq \max_{a \in \mathcal{A}} \mathcal{D}_f(P_a\|Q_a^\star) - \delta.$$

$\square$

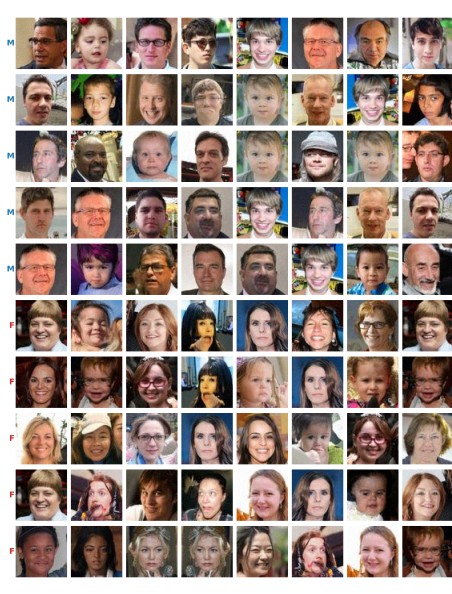 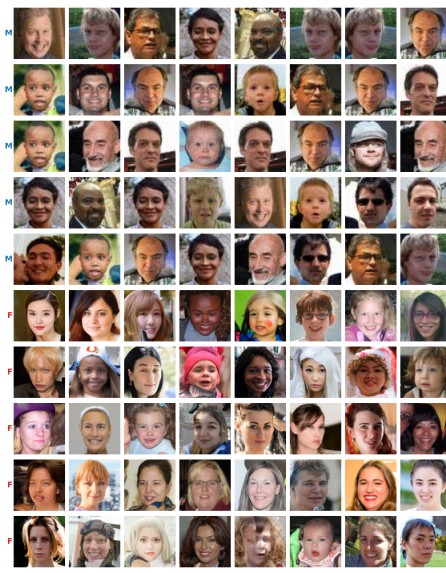

(a) Example 1            (b) Example 2

Figure 6: Samples from distribution of FFHQ faces with similar precision (79.06 (6a) and 77.12 (6b)) and similar recall $R$ (60.02 (6a) and 60.31 (6b)). However, the precision and recall for the sub-group are very different. In particular, in Example 1 the models generates slighly noised images for both classes. In Example 2, the model generates limited diversity and noisy images for Male Class while it generates high quality images and diverse samples for Female.

## B EXPERIMENTAL SETUP

### B.1 BUILDING ILLUSTRATIVE EXAMPLES

In this section, we provide additional details on the illustrative examples presented in Figure 1 in the main text. We construct two distinct set of examples to highlight the limitations of existing fairness metrics in generative models. Both examples are based on the FFHQ dataset (Karras et al., 2022), which contains high-quality images of human faces along with associated metadata, including gender information. For both examples, we used images of size $64 \times 64$ pixels generated using a pre-trained unconditional diffusion model EDM-VP (Karras et al., 2022). We first made sure that the proportion of male and female images in the set of generated images were equal to enforce MGO. Then to artificially create disparities in generation quality between the two groups, we manually applied random gaussian noise to the images of one of both groups. In Example 1, we added slight noise to images of both groups, resulting in a small but noticeable difference in quality between the original images and the ones in Example 1 (around $95\%$ precision dropping to $70\%$ precision for both groups). In Example 2, we added significant noise to the images of Male individuals and kept the images of Female individuals unchanged, resulting in a large disparity in quality between the two groups. To artificially reduce the diversity of the generated images we have duplicated on subset of images in both groups and applied random horizontal flips. In Example 1, we duplicated $20\%$ of the images in both groups, resulting in a small but noticeable difference in diversity between the two groups (around $60\%$ recall dropping to $45\%$ recall for both groups). In Example 2, we kept the Female images unchanged and duplicated 20 images of Male individuals, resulting in a large disparity in diversity between the two groups. The final precision and recall values for both examples are reported in Table 3. As shown in Figure 6 and Table 3, both examples have similar global precision and recall values, but the subgroup metrics differ significantly, illustrating the limitations of existing fairness metrics in capturing disparities in generation quality between sensitive groups.

Table 3: precision and recall for the two examples shown in Figure 6. The global precision and recall are similar, but the subgroup metrics differ significantly.

| Subset | Precision | Recall | M Recall | F Recall | M Precision | F Precision |
|--------|-----------|--------|----------|----------|-------------|-------------|
| Example 1 | 79.06 | 60.02 | 47.96 | 45.01 | 73.96 | 71.88 |
| Example 2 | 77.12 | 60.31 | 1.52 | 91.22 | 66.53 | 95.85 |

## B.2 RELATED WORKS AND OUR METHOD

In unconditional training, the model is simply optimized on the full dataset. Because the training proportions mirror those of the dataset, such models implicitly satisfy Matching Generative Odds (MGO) in expectation. However, they cannot correct for disparities beyond dataset proportions.

Conditional models, in contrast, allow us to explicitly control the proportions at generation time. By sampling the conditioning attributes in prescribed ratios, one can directly enforce both MGO and Equalized Generative Odds (EGO).

Rejection sampling (RS) provides another mechanism: generated samples are filtered until the desired proportions are reached. This method requires access to an oracle classifier (detailed in the diffusion and LLM sections). Its main advantage is the ability to enforce both MGO and EGO without retraining. However, RS can be extremely inefficient. For example, in the Wikipedia Biographies dataset where pretrained LLMs underrepresent female biographies (down to about 16%), achieving EGO may require generating more than five times as many samples. For this reason, we only applied RS to diffusion models, not to LLMs.

Table 4: Methods for generative fairness. The table indicates whether the method applies at training or inference time, and whether it can enforce Matching Generative Odds (MGO) and/or Equalized Generative Odds (EGO), and Equalized Generative Treatment (EGT).

| Method | Applies on | MGO | EGO | EGT |
|--------|-----------|-----|-----|-----|
| Standard Unconditional | Training | ✗ | ✗ | ✗ |
| Rejection Sampling (Zameshina et al., 2022) | Inference | ✓ | ✓ | ✗ |
| Standard Conditional | Training | ✓ | ✓ | ✗ |
| Reweighting (Choi et al., 2020) | Training | ✗ | ✓ | ✗ |
| Minimax Unconditional | Training | ✗ | ✗ | ✓ |
| Minimax Conditional | Training | ✓ | ✓ | ✓ |

Reweighting, introduced by Choi et al. (2020), adjusts the loss function according to dataset proportions. Concretely, the per-sample loss is rescaled by the inverse of the group frequency, thereby rebalancing the target distribution and encouraging the model to approach EGO.

Finally, Min–Max training directly targets Equalized Generative Treatment (EGT). Here, the objective is to minimize the maximum group-wise $f$-divergence, thereby aligning conditional qualities across groups. Both unconditional and conditional models can be trained in this manner.

We summarize the methods in Table 4 and give a pseudocode description of Min–Max training below.

---

**Algorithm 1:** Min–Max training with EMA of group-wise losses

1 Initialize parameters $\theta$ of model $Q_\theta$
2 Initialize moving averages $L_a$ for each group $a \in \mathcal{A}$
3 **for** *each training iteration* **do**
4      Sample minibatch with group labels $a$
5      Compute per-group losses $\ell_a$ (e.g., denoising loss or language modeling loss)
6      Update moving averages: $L_a \leftarrow \alpha L_a + (1-\alpha)\ell_a$
7      Identify worst group $a^* = \arg\max_a L_a$
8      Compute gradient of loss $\ell_{a^*}$ w.r.t. group $a^*$ only
9      Update parameters $\theta$ using this gradient

---

### B.3 DIFFUSION MODELS

#### B.3.1 EXPERIMENTAL SET-UP

We use the publicly available codebase of Karras et al. (2022) to train and evaluate diffusion models. We experiment with two different architectures: EDM-VP and EDM-VE. Both architectures are based on a U-Net architecture with attention layers, but they differ in the noise schedule and the parameterization of the denoising model. We refer the reader to Karras et al. (2022) for more details on the architectures. We have used the pre-trained models provided by Karras et al. (2022) as our baseline models for EDM-VP and EDM-VE on the dataset FFHQ dataset in our experiments.

#### B.3.2 TRAINING DIFFUSION MODELS

Training diffusion models consists in defining a sequence of noise levels $\sigma$ and minimizing a weighted sum of denoising objectives. Concretely, we introduce a denoising model $F_\theta(x, \sigma)$ and optimize

$$\mathcal{L}_{\text{Unconditional}}(\theta) = \mathbb{E}_{x \sim P, \sigma}\left[w(\sigma)\left\|F_\theta(x, \sigma) - x\right\|^2\right], \tag{8}$$

where $w(\sigma)$ are predefined weights. This objective encourages the model to predict the clean signal from its noisy counterpart at various noise levels.

In conditional training, the same principle applies, but the denoising function is extended to incorporate the group attribute $a$:

$$\mathcal{L}_{\text{Conditional}}(\theta) = \mathbb{E}_{x \sim P, \sigma}\left[w(\sigma)\left\|F_\theta(x, \psi(x), \sigma) - x\right\|^2\right]. \tag{9}$$

This setup allows explicit conditioning on sensitive attributes. In practice, both unconditional and conditional models have a similar number of parameters, with conditional models requiring slightly more due to the embedding of attribute information.

---

**Algorithm 2:** Min–Max diffusion training across noise levels

1 Initialize parameters $\theta$ of model $F_\theta$
2 Initialize moving averages $L_{a,\sigma}$ for each group $a$ and noise level $\sigma$
3 **for** *each training iteration* **do**
4     Sample minibatch of data points $x$ with group labels $a$ and multiple noise levels $\sigma$
5     Compute per-group, per-noise losses $\ell_{a,\sigma} = \|F_\theta(x, a, \sigma) - x\|^2$
6     Update EMA: $L_{a,\sigma} \leftarrow \alpha L_{a,\sigma} + (1 - \alpha)\ell_{a,\sigma}$
7     For each noise level $\sigma$, identify the worst group $a^*(\sigma) = \arg\max_a L_{a,\sigma}$
8     Collect losses $\ell_{a^*(\sigma),\sigma}$ across all $\sigma$
9     Compute gradient of the sum of these worst-case losses
10     Update parameters $\theta$ using this gradient

---

**Training details.** We train all diffusion models for 1M iterations with a batch size of 256 using the Adam optimizer with a learning rate of 1e-4. For Min–Max training, we use an exponential moving average (EMA) with a decay factor of 0.9 to track group-wise losses across noise levels. Training is performed on 8 NVIDIA v100 GPUs with 32GB of memory and takes approximately 5 hours for each model.

**Results** Across these tables, the values are normalized so that the lowest loss equals 1. We observe that conditional training helps flatten the loss landscape across groups and noise levels. Min–Max training reduces the gap between groups, though less strongly than conditional training. Reweighting also narrows disparities, in some cases even more effectively than Min–Max.

Table 6: Training loss (MSE reconstruction error) for different diffusion models on FFHQ dataset (conditional and unconditional). The values are averaged over all noise levels and classes. The loss is normalized with respect to the class with the lowest loss.

| Model | FFHQ (uncond.) | | | FFHQ (cond.) | | |
|---|---|---|---|---|---|---|
| | All | Male | Female | All | Male | Female |
| EDM-VP (Karras et al., 2022) | 2.29±0.80 | 3.88±1.80 | **1.00±0.00** | 1.02±0.02 | **1.00±0.00** | 1.03±0.03 |
| EDM-VE (Karras et al., 2022) | 1.02±0.02 | **1.00±0.00** | 1.04±0.03 | 1.02±0.01 | **1.00±0.00** | 1.03±0.02 |

Table 7: Training loss (MSE reconstruction error) for Min–Max diffusion models on FFHQ dataset (conditional and unconditional). The values are averaged over all noise levels and classes. The loss is normalized with respect to the class with the lowest loss.

| Model | FFHQ (uncond.) | | | FFHQ (cond.) | | |
|---|---|---|---|---|---|---|
| | All | Male | Female | All | Male | Female |
| EDM-VP (Karras et al., 2022) | 1.60±0.52 | 2.35±1.17 | **1.00±0.00** | 1.02±0.01 | **1.00±0.00** | 1.03±0.02 |
| EDM-VE (Karras et al., 2022) | 1.02±0.01 | **1.00±0.00** | 1.03±0.02 | 1.02±0.01 | **1.00±0.00** | 1.03±0.02 |

Table 8: Training loss (MSE reconstruction error) for Reweighting diffusion models on FFHQ dataset (conditional and unconditional). The values are averaged over all noise levels and classes. The loss is normalized with respect to the class with the lowest loss.

| Model | FFHQ (uncond.) | | | FFHQ (cond.) | | |
|---|---|---|---|---|---|---|
| | All | Male | Female | All | Male | Female |
| EDM-VP (Karras et al., 2022) | 1.24±0.15 | 1.53±0.40 | **1.00±0.00** | 1.02±0.01 | **1.00±0.00** | 1.04±0.02 |
| EDM-VE (Karras et al., 2022) | 1.03±0.01 | 1.05±0.03 | **1.00±0.00** | 1.02±0.01 | 1.03±0.02 | **1.00±0.00** |

In Figure 7, we plot the estimated loss for the different methods across noise levels for EDM-VE models. We observe that baseling models already have relatively balanced losses (compared to EDM-VP models in Figure 5 in the main text).

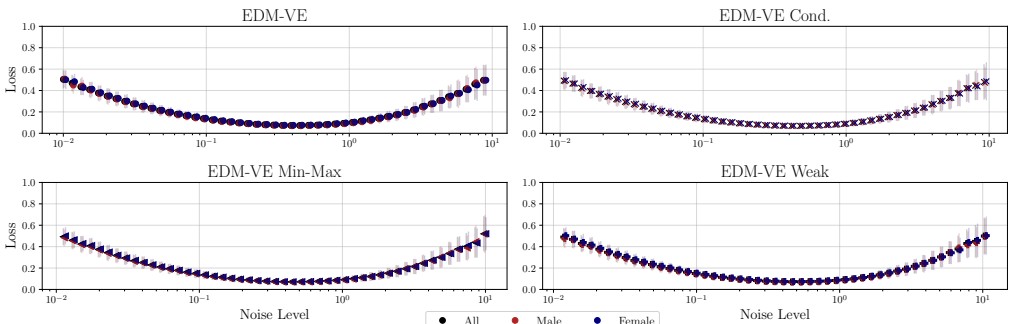

Figure 7: Comparison of the estimated loss for the baseline, Min-Max, conditional and reweighted methods on the FFHQ dataset with VP model. The loss is plotted for every noise level.

### B.3.3 EVALUATING DIFFUSION MODELS

We evaluate fidelity and diversity with the precision/recall (P&R) framework of Kynkäänniemi et al. (2019), while adopting the robust support–estimation procedure of Topological precision & recall (TopP&R) by Kim et al. (2023). In the P&R formalism, let $\mathcal{S}_{\text{real}}$ and $\mathcal{S}_{\text{gen}}$ be estimates of the supports of the real and generated feature distributions, respectively. *precision* is the fraction of generated features lying in $\mathcal{S}_{\text{real}}$ (sample fidelity), and *recall* is the fraction of real features lying in $\mathcal{S}_{\text{gen}}$ (coverage). This cleanly decouples realism from diversity and is the basis for our subgroup analyses.

**Topological support estimation (TopP&R).** Instead of $k$NN radii (as in Kynkäänniemi et al. (2019)), we estimate supports using the topological pipeline of Kim et al. (2023): persistent homology is used to retain only topologically and statistically significant structures (with confidence control), producing support sets that are more robust to outliers, hubness, and non-i.i.d. perturbations. P&R are then computed by testing membership of generated (resp. real) features in the topologically pruned support of the other set.

**Feature space and oracle.** To detect subtle distinctions in FFHQ, we extract features with a DINOv2 ViT (Oquab et al., 2024) fine-tuned on FFHQ. The same fine-tuned DINOv2 serves as an *oracle* to assign sensitive-attribute labels (male/female) to generated images, which we need for group-wise metrics.

**Random projections and repeated estimates.** For computational stability and speed, we apply a random Gaussian projection to a lower-dimensional subspace before running TopP&R. We repeat the entire TopP&R computation **10** times with independent projections and report the mean (and, when relevant, the standard deviation) of precision, recall, and $P+R$.

**Sample sizes.** Unless noted otherwise, we evaluate on **20k generated samples per class** and use a matched number of real reference samples per class from FFHQ to avoid class-size bias.

**Additional metrics.** For completeness, we also report FID as a global quality indicator. While FID conflates fidelity and coverage, TopP&R-based group-wise P&R drive the fairness conclusions in the main text.

**Results.** We present here the full tables of results for all diffusion models trained on FFHQ dataset. Tables 9 and 10 report the metrics for unconditional models, while Tables 11 and 12 report the metrics for conditional models. The results are discussed in Section 5 in the main text.

Table 9: Evaluation metrics for different Unconditional EDM-VP models on FFHQ dataset.

| Metric | Class | Pretrained | | | RW | | | Min-Max | | |
| | | Unconst. | RS-MGO | RS-EGO | Unconst. | RS-MGO | RS-EGO | Unconst. | RS-MGO | RS-EGO |
| --- | --- | --- | --- | --- | --- | --- | --- | --- | --- | --- |
| | All | 97.71 | 98.57 | 97.16 | 96.6 | 97.49 | 97.09 | 98.0 | 97.51 | 97.38 |
| Precision | Male | 96.34 | 95.17 | 95.58 | 96.15 | 95.97 | 96.94 | 95.9 | 96.42 | 95.97 |
| | Female | 95.9 | 97.18 | 97.76 | 95.32 | 96.53 | 96.63 | 95.65 | 96.83 | 95.93 |
| | All | 87.97 | 86.07 | 89.85 | 86.89 | 84.3 | 85.52 | 86.59 | 89.46 | 87.18 |
| Recall | Male | 89.61 | 91.73 | 91.68 | 87.44 | 87.9 | 86.54 | 91.44 | 89.2 | 91.11 |
| | Female | 90.96 | 87.5 | 87.21 | 88.51 | 86.59 | 86.31 | 91.23 | 87.62 | 91.45 |
| | All | 2.365 | 2.329 | 2.496 | 2.567 | 2.512 | 2.658 | 2.438 | 2.347 | 2.538 |
| FID | Male | 3.19 | 3.12 | 2.989 | 3.207 | 3.188 | 3.083 | 3.14 | 2.971 | 2.85 |
| | Female | 2.82 | 2.777 | 2.777 | 3.135 | 3.063 | 3.1 | 2.866 | 2.908 | 2.907 |
| | All | - | - | - | - | - | - | - | - | - |
| $\pi_a$ | Male | 0.442 | 0.445 | 0.498 | 0.461 | 0.449 | 0.499 | 0.348 | 0.446 | 0.499 |
| | Female | 0.558 | 0.555 | 0.502 | 0.539 | 0.551 | 0.501 | 0.652 | 0.554 | 0.501 |

Table 10: Evaluation metrics for different Unconditional EDM-VE models on FFHQ dataset.

| Metric | Class | Pretrained | | | RW | | | Min-Max | | |
| | | Unconst. | RS-MGO | RS-EGO | Unconst. | RS-MGO | RS-EGO | Unconst. | RS-MGO | RS-EGO |
| --- | --- | --- | --- | --- | --- | --- | --- | --- | --- | --- |
| | All | 97.71 | 98.57 | 97.16 | 96.6 | 97.49 | 97.09 | 98.0 | 97.51 | 97.38 |
| Precision | Male | 96.34 | 95.17 | 95.58 | 96.15 | 95.97 | 96.94 | 95.9 | 96.42 | 95.97 |
| | Female | 95.9 | 97.18 | 97.76 | 95.32 | 96.53 | 96.63 | 95.65 | 96.83 | 95.93 |
| | All | 87.97 | 86.07 | 89.85 | 86.89 | 84.3 | 85.52 | 86.59 | 89.46 | 87.18 |
| Recall | Male | 89.61 | 91.73 | 91.68 | 87.44 | 87.9 | 86.54 | 91.44 | 89.2 | 91.11 |
| | Female | 90.96 | 87.5 | 87.21 | 88.51 | 86.59 | 86.31 | 91.23 | 87.62 | 91.45 |
| | All | 2.365 | 2.329 | 2.496 | 2.567 | 2.512 | 2.658 | 2.438 | 2.347 | 2.538 |
| FID | Male | 3.19 | 3.12 | 2.989 | 3.207 | 3.188 | 3.083 | 3.14 | 2.971 | 2.85 |
| | Female | 2.82 | 2.777 | 2.777 | 3.135 | 3.063 | 3.1 | 2.866 | 2.908 | 2.907 |
| | All | - | - | - | - | - | - | - | - | - |
| $\pi_a$ | Male | 0.442 | 0.445 | 0.498 | 0.461 | 0.449 | 0.499 | 0.348 | 0.446 | 0.499 |
| | Female | 0.558 | 0.555 | 0.502 | 0.539 | 0.551 | 0.501 | 0.652 | 0.554 | 0.501 |

Table 11: Evaluation metrics for different Conditional EDM-VP models on FFHQ dataset.

| Metric | Class | Pretrained | | RW | | Min-Max | |
|---|---|---|---|---|---|---|---|
| | | RS-MGO | RS-EGO | RS-MGO | RS-EGO | RS-MGO | RS-EGO |
| Precision | All | 99.08 | 99.09 | 97.98 | 98.47 | 97.86 | 98.84 |
| | Male | 96.78 | 96.44 | 96.0 | 96.08 | 96.07 | 96.72 |
| | Female | 98.4 | 98.16 | 97.25 | 97.01 | 97.56 | 97.48 |
| Recall | All | 84.88 | 84.19 | 87.36 | 83.94 | 88.81 | 85.41 |
| | Male | 91.47 | 92.3 | 91.95 | 91.42 | 92.88 | 92.51 |
| | Female | 84.66 | 84.56 | 86.02 | 86.4 | 87.19 | 85.95 |
| FID | All | 2.754 | 2.846 | 2.557 | 2.647 | 2.596 | 2.623 |
| | Male | 3.405 | 3.362 | 3.284 | 3.231 | 3.466 | 3.4 |
| | Female | 3.322 | 3.366 | 3.024 | 3.063 | 2.987 | 3.016 |
| $\pi_a$ | All | - | - | - | - | - | - |
| | Male | 0.445 | 0.497 | 0.445 | 0.497 | 0.445 | 0.497 |
| | Female | 0.555 | 0.503 | 0.555 | 0.503 | 0.555 | 0.503 |

Table 12: Evaluation metrics for different Conditional EDM-VE models on FFHQ dataset.

| Metric | Class | Pretrained | | RW | | Min-Max | |
|---|---|---|---|---|---|---|---|
| | | RS-MGO | RS-EGO | RS-MGO | RS-EGO | RS-MGO | RS-EGO |
| Precision | All | 98.9 | 98.98 | 97.94 | 97.95 | 98.77 | 98.7 |
| | Male | 98.25 | 95.53 | 96.0 | 94.41 | 96.6 | 96.1 |
| | Female | 97.37 | 98.74 | 96.99 | 96.86 | 97.49 | 97.75 |
| Recall | All | 81.8 | 82.3 | 86.62 | 85.29 | 85.5 | 84.37 |
| | Male | 87.1 | 90.91 | 90.4 | 92.96 | 91.65 | 92.27 |
| | Female | 84.44 | 81.19 | 86.9 | 87.17 | 86.86 | 85.97 |
| FID | All | 2.88 | 3.0 | 2.521 | 2.604 | 2.865 | 2.897 |
| | Male | 3.545 | 3.501 | 3.301 | 3.237 | 3.871 | 3.802 |
| | Female | 3.433 | 3.475 | 2.931 | 2.971 | 3.157 | 3.201 |
| $\pi_a$ | All | - | - | - | - | - | - |
| | Male | 0.445 | 0.497 | 0.445 | 0.497 | 0.445 | 0.497 |
| | Female | 0.555 | 0.503 | 0.555 | 0.503 | 0.555 | 0.503 |

## B.4 LARGE LANGUAGE MODELS

### B.4.1 TRAINING LARGE LANGUAGE MODELS

We fine-tune the chat variants of LLaMA 3.2 at two scales (**1B** and **3B** parameters) using parameter-efficient adapters (LoRA). Intuitively, LoRA inserts small trainable matrices into the attention/MLP layers so that we only update a tiny fraction of weights, which is more memory- and time-efficient than full fine-tuning.

**Instruction format (unconditional).** We use a simple two-message chat template with a system instruction that specifies the desired style and a user query asking for a Wikipedia-style summary. The model is trained to produce the response.

```
SYSTEM_PROMPT = """
You are a helpful assistant that writes Wikipedia-style biography
summaries of notable individuals.
Your writing should follow the tone, style, and structure of
Wikipedia "Summary" sections:
- Neutral, encyclopedic tone
- No personal opinions or promotional language
- Concise but informative, focusing on key life events, career,
and achievements
- Chronological ordering of major facts
- Use proper sentences, not bullet points
```

```
- The biography text should start with "Biography:" followed
directly by the text in the next line.
Do not invent references or citations. Do not include external
commentary about the writing task.
"""

CHAT_ZERO_SHOT = (
    'Write the "Summary" section of a Wikipedia article about
    a person of your choice.'
)

messages = [
    {"role": "system", "content": SYSTEM_PROMPT},
    {"role": "user",   "content": CHAT_ZERO_SHOT},
]
```

**Conditional instruction format.**    To train *conditional* models, we keep the same template and add a short constraint specifying the sensitive attribute (here, gender) directly in the user message:

```
CHAT_ZERO_SHOT = (
    'Write the "Summary" section of a Wikipedia article about
    a person of your choice.'
)
if gender is not None:
    user_prompt = CHAT_ZERO_SHOT + f" It should be a biography
    of a {gender} person."
messages = [
    {"role": "system", "content": SYSTEM_PROMPT},
    {"role": "user",   "content": user_prompt},
]
```

This purely prompt-based conditioning makes it easy to control the proportion of generated attributes at inference time (e.g., sample 50/50 or 75/25), and is directly compatible with the EGO/MGO controls discussed earlier.

**Training details.**    We fine-tune the models for **6 epochs** with a batch size of **64** and a learning rate of **1e-4** using the AdamW optimizer. We use a LoRA rank of **8**. Training is performed on a single NVIDIA H100 GPU with 80GB of memory and takes approximately **2 hours** for the 1B model and **5 hours** for the 3B model.

### B.4.2   EVALUATING LARGE LANGUAGE MODELS

We evaluate text generations with the same P&R formalism and TopP&R support estimation used for images, but in a *text-embedding space*. Concretely, we embed both real (Wikipedia) and generated biographies with a strong sentence-embedding model (we use a 4B-parameter open-source text-embedding model; reference omitted for anonymity). We then apply a random Gaussian projection to reduce dimensionality and run TopP&R to estimate topological supports and compute precision/recall. This process is repeated **10** times with independent projections; we report the mean (and, when relevant, the standard deviation).

**Oracle for sensitive attribute.**    For LLM outputs, we assign gender labels using a simple pronoun-based heuristic: we scan each biography and count occurrences of gendered terms (e.g., *he/him/his* vs. *she/her/hers*). The majority class is used as the predicted label; ties or ambiguous cases are left unassigned. While noisy, this heuristic works well on short Wikipedia-style summaries and avoids extra training.

**Notes.**    As with images, we compute group-wise P&R (and P+R) under TopP&R on embeddings to quantify fidelity and coverage per sensitive group. We also report FID-like global metrics for completeness, but our fairness conclusions rely on group-wise TopP&R.

Table 13: Evaluation metrics for different Unconditional LLMs on Wikipedia Bio dataset.

| Metric | Class | Pretrained | | RW | | Min-Max | |
|---|---|---|---|---|---|---|---|
| | | 3b | 1b | 3b | 1b | 3b | 1b |
| Precision | All | 65.75 | 66.27 | 66.19 | 66.08 | 65.91 | 64.56 |
| | Male | 53.06 | 55.68 | 54.75 | 55.43 | 54.78 | 54.79 |
| | Female | 78.43 | 76.86 | 77.63 | 76.72 | 77.05 | 74.34 |
| Recall | All | 79.87 | 71.30 | 78.79 | 71.52 | 79.33 | 73.25 |
| | Male | 87.40 | 77.03 | 84.94 | 76.12 | 85.76 | 77.11 |
| | Female | 72.34 | 65.57 | 72.65 | 66.91 | 72.89 | 69.39 |
| $\pi_a$ | All | - | - | - | - | - | - |
| | Male | 82.67 | 81.44 | 83.54 | 81.60 | 83.74 | 81.49 |
| | Female | 17.33 | 18.56 | 16.46 | 18.40 | 16.26 | 18.51 |

Table 14: Evaluation metrics for different Conditional LLMs on Wikipedia Bio dataset.

| Metric | Class | Pretrained | | RW | | Min-Max | |
|---|---|---|---|---|---|---|---|
| | | 3b | 1b | 3b | 1b | 3b | 1b |
| Precision | All | 65.29 | 65.22 | 66.06 | 65.15 | 65.69 | 64.33 |
| | Male | 53.14 | 55.68 | 55.09 | 54.80 | 52.72 | 54.15 |
| | Female | 77.44 | 74.76 | 77.03 | 75.50 | 78.67 | 74.52 |
| Recall | All | 79.25 | 70.27 | 78.44 | 70.16 | 78.81 | 71.21 |
| | Male | 85.38 | 74.18 | 83.28 | 75.70 | 85.15 | 75.73 |
| | Female | 73.11 | 66.36 | 73.59 | 64.63 | 72.48 | 66.68 |
| $\pi_a$ | All | - | - | - | - | - | - |
| | Male | 50.00 | 50.00 | 50.00 | 50.00 | 50.00 | 50.00 |
| | Female | 50.00 | 50.00 | 50.00 | 50.00 | 50.00 | 50.00 |

### B.4.3 WIKIPEDIA BIOGRAPHIES DATASET

The Wikipedia Biographies dataset is derived from the Wikipedia Biographies dataset introduced by Bronnec et al. (2024), which contains header of biographies of individuals. Following, we present examples of biographies generated by the different models.

**Unconditional Generation:**

| Model | Method | Text | Gender |
|---|---|---|---|
| 1b | Pretrained | Joseph Louis Casimir Czerniak (Polish: [jzef luiz ksnak]; 15 February 1898 14 March 1943) was a Polish politician who served as the 20th Prime Minister of Poland from 1925 to 1928 and again from 1935 to 1939. He was the leader of the Polish Socialist Party, a Polish political party founded in 1919, and served as its leader from 1924 to 1935. He was the first Polish Prime Minister to be elected to the Polish Sejm. He was assassinated in 1943. Czerniak was born in Warsaw. He studied at the Warsaw University, graduating in 1921. He was a member of the Polish Socialist Party and participated in the PolishSoviet War. In 1925, he was appointed Prime Minister after the resignation of Stanisaw Wojciech Mikoaj Gombrzyski. He was re-elected in 1928, but was dismissed by the Sejm in 1935. He was assassinated in 1943. | male |

| | | | |
|---|---|---|---|
| 1b | Pretrained | Robert Charles Stevenson (29 June 1847 14 September 1927) was an English actor, comedian and comedian who appeared in numerous pantomimes and musical comedies, and was best known for his starring roles in the 1912 film "The Tragedy of Esther Waters" and the 1915 film "The Homecoming of Robin Hood". He was born in London and began his career as a professional actor in the West End, and later moved to the United States, where he appeared in a number of stage shows and films. He died in 1927. | male |
| 1b | Min-Max | Joseph Louis Broussard (November 19, 1881 July 8, 1962) was a Major League Baseball pitcher who played for the St. Louis Cardinals, Philadelphia Athletics, New York Yankees, and Chicago Cubs. He was a member of the Cardinals' 1922 World Series championship team and was a member of the 1926 National League pennant winning team. He was the first pitcher to throw a no-hitter in the National League, and was the first pitcher to throw a complete game shutout in the National League, accomplishing the feat in 1928 while pitching for the Yankees. Born in El Paso, Texas, Broussard began his baseball career with the Tucson Toros of the Pacific Coast League. He was signed by the Cardinals in 1909 and made his major league debut in 1910. He was traded to the Athletics in 1912, where he was released after the 1914 season. He returned to the Cardinals in 1915 and pitched for the team until his retirement in 1925. Broussard was inducted into the National Baseball Hall of Fame in 1962. | male |
| 1b | Min-Max | Robert "Bob" C. Henningsen (born June 7, 1935) is an American retired professional baseball player who played in Major League Baseball for 16 seasons as a catcher. He played his entire career with the St. Louis Cardinals, and is a member of the Cardinals' franchise's Hall of Fame. Henningsen was a key player on the Cardinals' 1962 World Series championship team and was a three-time All-Star. He was a member of the Cardinals' 1963 National League pennant team. He also won a World Series championship in 1964 with the Cardinals and was named the World Series MVP. | male |
| 1b | RW | Joseph Louis Casimir Czerniak (Polish: [jzef luiz ksnak]; 15 February 1898 14 March 1943) was a Polish politician who served as the 20th Prime Minister of Poland from 1925 to 1928 and again from 1935 to 1939. He was the leader of the Polish Socialist Party, a Polish political party founded in 1919, and served as its leader from 1924 to 1935. After the PolishSoviet War, he became a key figure in the Polish government and was appointed Prime Minister in 1935. He was the first Polish government leader to be imprisoned by the Nazis. He died in 1943 while in custody. Czerniak was born in the city of Krakw. He studied law at the University of Krakw, where he became involved in the Polish Socialist Party. He was a member of the Polish government in exile from 1925 to 1928, and was a key figure in the Polish government of 1935. He was imprisoned by the Nazis after the invasion of Poland in 1939 and died in custody in 1943. | male |
| 1b | RW | Robert Charles Stevenson (29 November 1847 14 September 1927) was an English actor, comedian and comedian who appeared in numerous pantomimes and musical comedies, and was best known for his starring roles in the 1912 film "The Tragedy of Esther Waters" and the 1915 film "The Homecoming of Robin Hood". He was born in London and educated at Harrow School and the Royal Academy of Dramatic Art. He began his acting career in London's West End, performing in a number of pantomimes and musical comedies, before moving to the United States, where he became a popular actor and comedian in the early 20th century. He appeared in a number of films, and toured the United States in the 1910s and 1920s, performing in vaudeville and revues. He died in 1927. | male |

| 3b | Pretrained | Joseph "Joe" John "Bull" Connolly (March 2, 1895 November 7, 1969) was an American professional baseball player who played in Major League Baseball (MLB) for 17 seasons, all with the New York Yankees. He was a left-handed pitcher and a member of the Yankees' 1923 World Series championship team. Connolly was the Yankees' ace pitcher in the 1923 season, posting a 207 winloss record with a 2.61 earned run average (ERA). He was also a key member of the Yankees' 1927 team that won the World Series, and pitched in 18 games for the team. Connolly was a member of the Yankees' rotation in the early 1920s, but his career was marked by inconsistency and a series of injuries. He was a member of the Yankees' starting rotation for only 18 of his 17 seasons in the major leagues, and he was out of baseball for several years due to various injuries. He was released by the Yankees in 1930, but returned to the team in 1934. He played for the Yankees until 1938, when he retired. | male |
| 3b | Pretrained | Charles Charles Victor Sargent (March 21, 1841 May 25, 1927) was an American landscape gardener, horticulturist, and botanist. He was a professor of horticulture at the Massachusetts Agricultural College (now the University of Massachusetts Amherst) for over 40 years and was the first president of the American Horticultural Society. He is best known for his work on the development of the Boston Public Garden and the Boston botanical garden at Harvard University. Sargent was born in Boston, Massachusetts, and grew up in a family of modest means. He was apprenticed to a nurseryman at the age of 15, and after completing his apprenticeship, he worked in several other gardens in the United States. In 1872, he was appointed as a professor of horticulture at the Massachusetts Agricultural College, and remained there until his retirement in 1913. Sargent was a prolific writer, and his books on horticulture were widely read in the United States and abroad. He was also a prolific writer on botanical topics, and his works were published in numerous journals and magazines. | male |
| 3b | Min-Max | Joseph "Jack" Gorman (born 10 November 1985) is a former English professional footballer who played as a striker for several clubs, including Wigan Athletic, Ipswich Town and Crawley Town. Gorman began his career at Wigan Athletic, signing for the club in 2005 from Manchester City. He made his debut for the club in 2005 and scored one goal in 14 appearances. He was released from the club at the end of the 200506 season and joined Ipswich Town in 2006. He spent two years with the club, scoring six goals in 51 appearances before moving to Crawley Town in 2008. He spent two seasons with the club, scoring 21 goals in 69 appearances before leaving the club in 2010. | male |
| 3b | Min-Max | Charles William Williams (6 July 1831 7 January 1910) was an English-born Australian cricketer who played first-class cricket for the New South Wales cricket team. He was a right-handed batsman and occasional wicket-keeper. Born in England, Williams moved to Australia with his family when he was eight years old. He was educated at the Sydney Church of England Grammar School and began his cricketing career with the school team, before making his first-class debut for New South Wales in 1853. He played for the team for the next 12 years, and was a member of the 185960 team that toured England, where he also played for the Marylebone Cricket Club. He was known for his batting, particularly his ability to score big centuries. He played in the first ever Test match in 1877 and was also a member of the New South Wales team that toured England in 188283. He retired from first-class cricket in 1865, but continued to play in minor league cricket until his death in 1910. | male |
| 3b | RW | Joseph "Joe" O'Connell (May 19, 1920 August 24, 1997) was an American politician who served as a member of the New Jersey General Assembly from the 2nd district from 1958 to 1964 and from 1967 to 1971. He was a Democrat. O'Connell was born in New York City and graduated from the New York City College of Pharmacy. He served in the United States Army during World War II and was awarded the Purple Heart. He graduated from Rutgers University and later earned a master's degree in education from Columbia University. He was a teacher and a pharmacist before entering politics. He was elected to the New Jersey General Assembly in 1957 and served until 1964. He was defeated for re-election in 1964, and did not seek election again until 1966. He served another term until 1971, when he was defeated in a Republican primary. | male |

| | | |
|---|---|---|
| 3b | RW | Charles Francis "Charlie" Jones (February 27, 1886 January 15, 1968) was an American baseball player and manager in Major League Baseball. He played in Major League Baseball from 1909 to 1925, playing for the Philadelphia Athletics, New York Giants, and Boston Braves. He also managed the St. Louis Browns from 1928 to 1929. Jones was a four-time All-Star and won the National League batting title in 1912. He is a member of the National Baseball Hall of Fame. He is the only player in Major League Baseball history to have won a batting title, a Gold Glove Award, and a World Series championship. | male |

## Conditional Generation:

| Model | Method | Text | Gender |
|---|---|---|---|
| 1b | Pretrained | Vivian Joseph O'Neal (born April 21, 1987) is an American professional baseball pitcher who played in Major League Baseball (MLB) as a relief pitcher for the New York Yankees. A former high school standout, he played college baseball for the University of Michigan Wolverines. He was drafted by the Yankees with the 8th round pick of the 2006 MLB draft. O'Neal was named the Yankees' Minor League Pitching Prospect of the Year in 2008. He was named a Golden Glove winner for his defensive play at shortstop in 2011. He was traded to the Los Angeles Angels in 2012. O'Neal was released by the Angels in 2013 and signed with the Baltimore Orioles in 2014. He was traded to the Kansas City Royals in 2015. He was traded to the Texas Rangers in 2016. He was released by the Rangers in 2017 and signed with the New York Yankees in 2018. He was traded to the Miami Marlins in 2019. He was released by the Marlins in 2020 and signed with the Seattle Mariners in 2021. | male |
| 1b | Pretrained | Lucius Julius Caesar (c. 52 BC 44 BC) was a Roman general and statesman who served as dictator of Rome in 49 BC. He was a member of the Julian clan and a member of the Julian clan, a patrician family that played a significant role in Roman politics. Caesar was the son of Gaius Julius Caesar and Cornelia, a member of the Julian clan. He was appointed as a quaestor in 43 BC, and served in the Roman Senate in 42 BC, but was expelled from the Senate in 41 BC after he was accused of treason and exiled to the island of Corsica. He returned to Rome in 40 BC, and served as a quaestor in 39 BC. He was appointed dictator of Rome in 49 BC, and was elected dictator for the fourth time in 44 BC, becoming the first Roman dictator since 49 BC. He was assassinated in 44 BC. Caesar's reign was marked by a series of military victories against the Gauls, and he was celebrated for his military campaigns, which included the conquest of the Roman province of Hispania, the conquest of the island of Sicily, and the expulsion of the Parthian Empire from the Roman province of Cappadocia. He also reformed the Roman calendar, introducing the Julian calendar, and reformed the Roman Senate, establishing the office of the praetor maximus. | male |
| 1b | Min-Max | Vivian James (ne Wilson; 4 October 1919 28 December 1994) was an English professional golfer who competed in the 1947 British Open and the 1951 US Open. She was the first woman to win a major championship. She was also the first woman to be ranked number one in the world golf rankings for a period of two years. She was a member of the first class of golfers to be inducted into the World Golf Hall of Fame. | female |
| 1b | Min-Max | Lucius Marcellus Valerius Rufus (fl. 1st century BC) was a Roman senator and statesman from the gens Rufi, a patrician family that included several prominent politicians and military leaders in the Roman Republic. He was a member of the first century BC, and was a member of the Curia of the Roman Senate from 41 BC until his death in 1 BC. Rufus was known for his political and military career in the late Republic, during which he served as a general under the dictator Julius Caesar. He was appointed consul in 41 BC and served as governor of the province of Asia in 39 BC. He died in 1 BC. | male |

| | | | |
|---|---|---|---|
| 1b | RW | Vivian Vess, ne (born 1956) is a Canadian actress, best known for her roles in the TV series The Shield and its spin-off series Alphas.  She was born in New York City and began her career in television and film in the early 1980s.  She began as a child actress on the soap opera The Young and the Restless and later appeared on the television series Perfect Strangers, The Love Boat, and The Golden Girls.  She gained recognition for her roles in the television series The Shield and its spin-off series Alphas, for which she won a Canadian Screen Award for Best Supporting Actress in a Drama Series. | female |
| 1b | RW | Lucius Quinctius Cincinnatus (c.  180c.  90 BCE) was a Roman statesman and military leader of the late Roman Republic.  He was a member of the noble family of the Cincii, and was a prominent figure in the late Roman Republic, serving as a censor, quaestor, and tribune of the plebs.  Cincinnatus was born in 180 BCE. He was a veteran of the Second Punic War, and served as a quaestor in the late 2nd century BCE. He was elected as a censors in 247 BCE, and then as a tribune of the plebs in 229 BCE. He served as censor again in 219 BCE, and was elected as a consul in 216 BCE. He was appointed as dictator in 214 BCE, and then as dictator again in 211 BCE. He was assassinated in 211 BCE, and succeeded by his nephew, Lucius Quinctius Calvus. | male |
| 3b | Pretrained | Joseph "Joe" O'Connell (May 19, 1920 August 4, 1997) was an American politician who served as a member of the New Jersey General Assembly from the 9th district from 1958 to 1964 and from 1967 to 1971.  He was a Democrat.  O'Connell was born in New York City and graduated from the New York City High School of Commerce.  He served in the United States Army during World War II and the Korean War, and worked as a laborer and a truck driver.  He was elected to the General Assembly in 1957 and served until 1964.  He was again elected in 1966 and served until 1971. | male |
| 3b | Pretrained | Robert Charles "Bob" Jones III (born March 30, 1940) is an American businessman, politician, and Christian minister.  He is best known for founding Liberty University, a Christian university in Lynchburg, Virginia, and for his advocacy of Christian nationalism and conservative politics.  He was the chairman of the Board of Directors of the Family Research Council, a conservative Christian advocacy group, from 1982 until 2012.  He was also the founder and chairman of the World Relief and Development Council, a Christian charity.  Jones founded Liberty University in 1971 as a small Bible college in Lynchburg, Virginia, and expanded it into a large university by the 1980s.  In the 1980s, he was a prominent figure in the Christian Right, and he ran for the Republican presidential nomination in 1988.  He has been a vocal critic of the separation of church and state and has advocated for the teaching of creationism in public schools.  He has been described as a "Christian fundamentalist" and a "evangelical". | male |
| 3b | Min-Max | Joseph "Joe" Jones (February 28, 1897 December 23, 1971) was an American professional baseball player who played for the Philadelphia Athletics, Chicago Cubs, and Brooklyn Dodgers of Major League Baseball (MLB). He played as a pitcher and outfielder.  Jones was born in New York City and attended George Washington University, where he played college baseball for the George Washington Pioneers.  After college, he was signed by the Athletics and made his MLB debut in 1918.  He played for the Athletics for eight seasons, including three All-Star appearances, and was a member of the 1927 World Series team.  He was traded to the Cubs in 1929, where he played for two seasons, and was then traded to the Dodgers in 1931.  He played for the Dodgers for four seasons before his MLB career was cut short by a series of injuries. | male |

| | | | | |
|---|---|---|---|---|
| 3b | Min-Max | Robert James "Bobby" Jones (23 July 1890 2 September 1971) was an English professional footballer who played as a centre-half. He made over 200 appearances for the first team of Southampton Football Club, and also played for the England national team. Born in Southampton, Jones began his career with local side East End United before joining Southampton in 1908. He made his first-team debut in 1910 and became a regular player for the club, helping them win the Southern League title in 191213. He also played for the England national team, earning 14 caps, and was part of the team that won the 1920 Summer Olympics. Jones moved to Fulham in 1914, but returned to Southampton in 1919, playing until his retirement in 1925. He later managed the club's youth team and was appointed chairman of the club in 1954. He was knighted in 1958 for his services to football. | male |
| 3b | RW | Joseph "Joe" Lawler (born 2 January 1968) is an English former professional footballer who played as a midfielder for several clubs including Manchester City, Ipswich Town and Sunderland. He also had a brief spell at Middlesbrough. Lawler began his career with his hometown club Manchester City, before moving to Ipswich Town in 1990. He won the First Division title in his first season at the club, and played in the 1992 FA Cup final. He moved to Sunderland in 1993, where he won the First Division title again, and played in the 1994 FA Cup final. He moved to Middlesbrough in 1995, but left after just one season. He had a brief spell at Norwich City in 1997 before retiring. | male |
| 3b | RW | Robert Francis "Bob" Johnson (January 23, 1894 August 7, 1971) was an American professional baseball player. He played in Major League Baseball (MLB) as a pitcher for the St. Louis Browns from 1913 to 1918. He was a left-handed thrower and batted and threw right-handed. Johnson was a member of the 1915 World Series championship team. He is best known for throwing a no-hitter in Game 1 of the 1915 World Series, and for his 1916 season in which he won 13 games and lost just two, and was named the American League leader in shutouts with 11. Johnson was also the American League leader in shutouts with 11 in 1917, and he was a member of the St. Louis Browns' 1918 World Series championship team. Johnson's career was cut short by an injury, as he suffered a shoulder injury in 1918 and was forced to retire from baseball. He later worked as a baseball scout for the Chicago Cubs and was also involved in the promotion of the St. Louis Browns. | male |