# OpenReview forum: "Equalized Generative Treatment: Matching $f$-divergences for Fairness in Generative Models"
_ICLR.cc/2026/Conference — Submitted to ICLR 2026_

### Official Review · Reviewer_hajo · 2025-10-29

**Soundness:** 2
**Presentation:** 2
**Contribution:** 1
**Rating:** 2
**Confidence:** 5

**Summary:**

This paper addresses fairness in generative models and argues that existing fairness notions are brittle, as they can equalize group proportions while still generating lower-quality samples for certain groups. To overcome this, the authors propose Equalized Generative Treatment (EGT), which enforces fairness by requiring parity of f-divergences between real and generated distributions across sensitive groups. They analyze its theoretical properties, highlight the inherent fairness–performance trade-off, and evaluate several training strategies (reweighting, min–max optimization, conditional training) on diffusion models and LLMs, showing reduced inter-group quality gaps without major loss in overall performance.

**Strengths:**

- The paper addresses an important topic: fairness in generative models.
- The writing is clear and easy to follow.
- Experiments cover various modalities of data: image and text.

**Weaknesses:**

- **Trivial central claim.** A large portion of the paper demonstrates that proportion-based fairness can still lead to quality disparity across groups, which is a natural and well-known issue. As demonstrated long ago (e.g., in [1]), the difference of intra-group FID already captures this discrepancy without defining new metrics.

- **Straightforward definition.** The proposed EGT metric is a direct restatement of equalizing f-divergences across groups, which is conceptually simple and lacks novelty.

- **No dedicated algorithm.** While the metric is defined, no new optimization method is introduced. Section 5 only reuses existing methods (reweighting, min–max, conditional training), and their relationship to EGT optimization remains unclear.

- **Outdated baseline methods.** The study heavily relies on older fairness strategies (e.g., Choi et al. 2020, classical min–max training [2,3]), limiting its relevance to modern generative frameworks.

- **Weak evaluation metrics.** Precision and recall are insufficient to capture generative capability in a thorough manner; stronger metrics such as density and coverage [4] would provide a more meaningful assessment.



----
**References**

[1] A Fair Generative Model Using LeCam Divergence, AAAI 2023.

[2] Minimax Group Fairness: Algorithms and Experiments, Arxiv 2020.

[3] Fairness without Demographics through Adversarially Reweighted Learning, NeurIPS 2020.

[4] Reliable Fidelity and Diversity Metrics for Generative Models, ICML 2020.

**Questions:**

See weaknesses.

---

> ### Author Response · Authors · 2025-11-21
>
> We thank the reviewer for his time. However, we respectfully disagree with the assessment made in the review and would like to clarify several points. We systematically address each of the concerns raised by the reviewer below. We hope that this raises the reviewer's concerns and helps them reassess the quality of our contribution.
>
> **On the alleged triviality of our main theoretical result.**
> We do not believe that manipulating surjectivity properties of f-divergences in probability measure spaces is trivial. Theorem 3.1 is not a restatement of a known fact but a non-obvious analytical result that required careful construction. It formalizes, for the first time, that *even under perfect proportion matching*, one can obtain *any* gap in conditional f-divergences. This is mathematically non-trivial and, to the best of our knowledge, absent from prior work.
>
> **Regarding the claim: “As demonstrated long ago (e.g. [15]), intra-group FID captures the issue.”**
> This is factually incorrect. While [15] empirically reports intra-class FID gaps, the paper does *not* study their relation to classical fairness criteria such as EGO or MGO, nor does it characterize incompatibilities between proportion-based metrics and quality-based ones. Their analysis focuses solely on FID, even though by 2023 many alternative metrics (e.g. Precision and Recall [16-19]) were already widely used Some references?.
> Our work (1) formalizes the theoretical relationship (or, more precisely, the *lack* of relationship) between quality metrics and fairness notions such as EGO/MGO; (2) extends the analysis to a broader class of f-divergences; and (3) demonstrates the same brittleness for text as well as images. These contributions are not present in [15].
>
> **“Straightforward definition” of EGT.**
> We view this simplicity as a strength. Metrics intended for fairness evaluation should be interpretable and easy to compute. While the *statement* of the definition is simple, the *implications* are not: the induced fairness–performance trade-offs, the conditional closure analysis, and the relation to existing fairness criteria are all novel and non-straightforward. The depth of the contribution lies in the analysis, not in producing a complicated formula.
>
> **On the absence of a new algorithm.**
> We explicitly state in our introduction that our goal is *not* to propose a new optimization method. The contributions of the paper are:
> – introducing a general and principled fairness metric;
> – analyzing its theoretical consequences;
> – empirically studying the effect of existing strategies on this definition.
> This is aligned with our stated objectives, and we do not claim algorithmic novelty.
>
> **Regarding “outdated baseline methods.”**
> If the reviewer believes more recent baselines exist, we kindly ask for explicit references. The three methods we evaluate are *not* outdated:
>
> - Reweighting is the standard approach in fairness-aware generative modeling and remains used in very recent works [1–5] (all in 2025).
> - Min–Max optimization is directly related to our Theorem 4.3 and remains central in modern fairness and robustness research, including very recent LLM work [6, 7].
> - Conditional training is theoretically connected to our brittleness result (Theorem 3.1).
>
> These choices are therefore well-grounded and relevant to the contribution.
>
> **“Weak evaluation metrics.”**
> We believe this would be a fair point *if* we were using the original precision/recall estimator of Kynkäanniemi et al. However, as pointed out by Reviewer hkXX, we use the more robust TopP&R estimator [8], specifically designed to address instability and outlier sensitivity.
> Density and Coverage are indeed available, but they are less widely used than PR ([16-20]) and, importantly, they do *not* fit naturally into the f-divergence framework underpinning EGT. Moreover, they are *not used at all* in the LLM literature, while PR-based metrics are employed routinely in recent works [9–14].

---

> > ### Author Response · Authors · 2025-11-21
> > ****References cited in this rebuttal****
> >
> > 1. FairBiNN: On Balancing Fairness and Accuracy via Stackelberg Equilibrium, Yazdani-Jahromi et al.
> > 2. FORML: Learning to Reweight Data for Fairness, Yan et al.
> > 3. PFGuard: A Generative Framework with Privacy and Fairness Safeguards, Kim et al.
> > 4. Benchmarking the Fairness of Image Upsampling Methods, Laszkiewicz et al.
> > 5. Balancing Act: Distribution-Guided Debiasing in Diffusion Models, Parihar et al.
> > 6. Distributionally Robust Language Modeling, Oren et al.
> > 7. Distributionally Robust Neural Networks for Group Shifts, Sagawa et al.
> > 8. TopP&R: Robust Support Estimation Approach for Evaluating Fidelity and Diversity, Kim et al.
> > 9. Benchmarking Linguistic Diversity of Large Language Models, Guo et al.
> > 10. Mind the Gap: Conformative Decoding to Improve Output Diversity, Peeperkorn et al.
> > 11. Evaluating List Construction and Temporal Understanding in LLMs, Dumitru et al.
> > 12. CLNX: Bridging Code and Natural Language for Vulnerability Detection, Qin et al.
> > 13. Multimodal LLMs in Construction Hazard Recognition, Chaudhary et al.
> > 14. Sustainability Identification in Stack Exchange Posts, Tudor-Mihai.
> > 15. A Fair Generative Model Using LeCam Divergence, Um et al.
> > 16. Consistency Models, Song et al.
> > 17. Diffusion-GAN: Training GANs with Diffusion, Wang et al.
> > 18. Flow Matching in Latent Space, Dao et al.
> > 19. StyleGAN-XL: Scaling StyleGAN to Large Diverse Datasets, Sauer et al.
> > 20. Diffusion Models Beat GANs on Image Synthesis, Dhariwal and Nichol.

---

> ### Comment · Reviewer_hajo · 2025-11-26
>
> Thank you for the rebuttal. I appreciate the clarifications provided. Below, I outline the remaining concerns that were not fully addressed in the authors’ response.
>
> **On the alleged triviality of our main theoretical result.**
>
> My point was that the idea behind the main claim is trivial and highly expected. I was not saying that the theoretical derivation itself is trivial. When applying existing fairness techniques to enforce perfect proportions (for example, 5:5 gender), it is very natural and obvious that quality unfairness across sensitive groups can still appear. This is the sense in which I called the claim trivial.
>
> **Regarding the claim “As demonstrated long ago (e.g. [15]), intra-group FID captures the issue.”**
>
> I agree that prior work did not provide the same theoretical analysis the authors present here. However, the statement *“As demonstrated long ago (e.g. [15])”* is not factually incorrect. In [1], the authors empirically demonstrate that quality degradation in minority groups appears as intra-group FID gaps. Since FID is essentially an earth-moving distance, the difference of intra-group FIDs naturally reflects the difference in earth-moving distance between the target and generated distributions across sensitive groups. The definition proposed in Definition 4.1 is essentially replacing this earth-moving distance with a general f-divergence. From this perspective, I am not convinced by the novelty of the proposed metric. Even if the novelty is limited, I might still find it meaningful if the paper offered algorithmic insight, but as I mentioned in my initial review above, that is also not the case.
>
>
> **“Straightforward definition” of EGT.**
>
> I agree that simplicity can be a strength. But the definition itself is still straightforward, as mentioned above.
>
>
> **On the absence of a new algorithm.**
>
> Since the definition is straightforward and simple, one would expect the paper to provide an algorithm that actually optimizes it in a meaningful way. For instance, Contribution 2 of Section 1 states:
>
> > *"We further show that applying this definition promotes the minimization of the highest f-divergence between conditionals among sesitive groups, which naturally leads us to study several optimization methods to improve fairness."*
>
> However, despite this claim that the definition *“promotes”* such an optimization, the paper does not offer any concrete method or intuition for how one might actually optimize the proposed metric. Instead, the authors only point out a connection to existing min-max training, without proposing a dedicated approach. As a result, I find the practical implication of the work very limited.
>
>
> **Regarding “outdated baseline methods.”**
>
> For instance, the authors could have included a more recent reweighting-based method such as [2], which provides improved results compared to [3]. In addition, many other recent approaches exist [4–8], and it is unclear why none of them were included or compared.
>
> **“Weak evaluation metrics.”**
>
> No additional concerns. I agree with the authors' rebuttal.
>
>
> ---
> **References**
>
> [1] A Fair Generative Model Using LeCam Divergence, AAAI 2023
>
> [2] Training unbiased diffusion models from biased dataset, ICLR 2024
>
> [3] Fair Generative Modeling via Weak Supervision, ICML 2020
>
> [4] FairBiNN: On Balancing Fairness and Accuracy via Stackelberg Equilibrium, Yazdani-Jahromi et al.
>
> [5] FORML: Learning to Reweight Data for Fairness, Yan et al.
>
> [6] PFGuard: A Generative Framework with Privacy and Fairness Safeguards, Kim et al.
>
> [7] Benchmarking the Fairness of Image Upsampling Methods, Laszkiewicz et al.
>
> [8] Balancing Act: Distribution-Guided Debiasing in Diffusion Models, Parihar et al.

---

> > ### Author Response · Authors · 2025-11-28
> >
> > **On the alleged triviality of our main theoretical result. & Regarding the claim “As demonstrated long ago (e.g. [15]), intra-group FID captures the issue.”**
> >
> > We agree that disparities are expected when enforcing proportion matching alone, but our result goes significantly further. As we will emphasize more clearly in the final version of the paper, we show that these disparities can in fact be made *arbitrarily large*: all the error may collapse onto a single attribute. This phenomenon does **not** necessarly hold for integral probability metrics (IPMs) such as the Earth Mover’s Distance or MMD.  To establish this, we prove two intermediate results of independent interest: Lemma A.1 on the surjectivity of f‑divergences on the interval $(0, f(0) + \bar f(+\infty))$, and Lemma A.2 on the linear decomposition of f‑divergences for mixtures with disjoint supports. Both results fail for IPMs, including Wasserstein distances. (IPMs do exhibit surjectivity, but only on a much more limited range entirely dependent on the target distribution.)  Thus, the statement that disparities can be *arbitrarily large* is strictly stronger than the observation that disparities merely exist. This behavior is specific to f‑divergences, not IPMs. Consequently, metrics based on f‑divergences—such as KL (for MLE in LLMs), Jensen–Shannon (for GANs), or Precision/Recall (for generative evaluation)—are inherently more prone to extreme inter‑group disparity than IPM‑based metrics such as FID. For this reason, we believe our Definition 4.1 is particularly relevant.
> >
> > **On the absence of a new algorithm.**
> >
> > Our goal is not to introduce a new optimization scheme, but to study the implications of EGT. In this regard, we show that min–max training is a theoretically grounded approach to reducing EGT, and that Theorem 4.3 guarantees optimal overall f‑divergence behavior under EGT constraints. In our view, this provides sufficient practical guidance without reinventing existing, well‑understood optimization paradigms.
> >
> > **On Regarding “outdated baseline methods.”**
> >
> > We thank the reviewer for pointing out additional references. In fact, the method from [2] is already included in our diffusion‑model experiments; it corresponds to a reweighting strategy dependent on the noise level. Regarding the other references:
> > - [4] relies on model‑class assumptions that do not hold for diffusion models or LLMs;
> > - [5] is unnecessary in our setting since we can compute weights directly;
> > - [6] is itself the same reweighting‑based fairness method and a method to enforce differentiable privacy;
> > - [7] is a benchmarking paper and does not introduce a new optimization method;
> > - [8] concerns sampling‑time adjustments only, not training techniques for diffusion models.
> > For these reasons, these approaches are not directly applicable or do not provide additional insights relative to the methods we already study.

---

### Official Review · Reviewer_hkXX · 2025-11-01

**Soundness:** 2
**Presentation:** 2
**Contribution:** 3
**Rating:** 6
**Confidence:** 4

**Summary:**

This paper studies fairness in generative models and argues that proportion-based criteria, e.g. equalized generative odds (EGO) and matching generative odds (MGO) can be satisfied even when generation quality differs substantially across demographic groups. The authors formalize this limitation and introduce Equalized Generative Treatment (EGT), which asks models to deliver comparable quality for each group by equalizing a standard measure of model-versus-data mismatch across groups. They show this effectively concentrates optimization on the worst-served group. They evaluate three practical training strategies (reweighting, a simple min-max objective, and conditional training) on diffusion models (FFHQ faces) and small LLMs (Wikipedia-style biographies), reporting reductions in per-group precision/recall gaps relative to odds-based baselines.

**Strengths:**

1. The theoretical result shows you can satisfy odds-based fairness and still be arbitrarily worse for one group. This cleanly formalizes a widely suspected issue.
2. The paper evaluates multiple model families (EDM-VP, EDM-VE, LLaMA-3.2) across different modalities (images, text) with multiple optimization strategies. The TopP&R framework for evaluation is more principled than standard kNN-based precision/recall.
3. Group-wise Topological Precision & Recall mitigates brittleness in support estimation and decouples fidelity from coverage. Repeated random projections add robustness.

**Weaknesses:**

1.  EGT is defined with standard divergences, but the paper does not provide a direct, ready-to-use training objective that enforces EGT. Experiments rely on proxies (reweighting, min-max, conditional training).
2. Table 1 shows conditional training achieves excellent  $\delta$-FID but significantly worse $\delta$-PR compared to unconditional methods. This directly contradicts the expectation that specific conditioning improves group-wise control. The paper mentions this result but provides no explanation or investigation.
3. Figure 4 clearly shows enforcing fairness raises all groups to worst-case performance, and Theorem 4.3 implies this mathematically.
4. Experiments focus mainly on a binary gender attribute and do not study intersectional groups, limiting external validity.
5. Most tables lack error bars.
6. Core evaluation details (TopP&R configuration, feature spaces/oracles, sample sizes), plus many training specifics and full tables, presented in the appendix, not the main paper.

**Questions:**

1. EGT is defined with a generic divergence family. Which one do you recommend for different modalities (images vs. text), and how sensitive are conclusions to this choice?
2. How do diffusion results change if you swap the FFHQ gender oracle (e.g., a different backbone/threshold)? How often do oracle disagreements flip group-gap conclusions?
3. Can EGT extend to multiple attributes and their intersections? How does this affect sample efficiency and numerical stability?

---

> ### Author Response · Authors · 2025-11-21
>
> We thank the reviewer for the careful reading and for the high-quality assessment of our paper, in particular regarding the evaluation methodology and the robustness offered by Topological Precision & Recall. We appreciate the clear expertise demonstrated in metric design and believe this helps contextualize our contribution in relation to recent advances.
>
> **Clarifying the scope of our contribution.**
> As for Reviewer 1, we emphasize that our goal is *not* to propose new fairness-improving algorithms. Instead, we aim to *analyze existing methods* (reweighting, min–max, and conditional training) through the lens of EGT. We acknowledge that the short discussion in the main paper (paragraph lines 390–401) does not sufficiently highlight their theoretical effects, and we will expand this section.
>
> - **Reweighting (RW)** is a classical, inexpensive approach to enforce EGO. It has no theoretical guarantee for EGT, but we included it because it is one of the standard baselines in the literature.
> - **Conditional training** disentangles the loss by class and avoids the pitfall highlighted in Theorem 3.1: a low global f-divergence hiding a large error concentrated on one class. Full independent training per class is unrealistic, so conditional models offer a practical middle ground.
> - **Min–Max training** is the *only* method among the three that explicitly enforces EGT at the level of the *training* divergence (though not necessarily the evaluation divergence), which is why it often yields the most consistent fairness improvements.
>
> **Regarding Table 1 and PR performance.**
> Conditional training does not necessarily improve EGT on PR, and this is precisely aligned with our theoretical message: controlling conditionals does not automatically reduce disparities in precision/recall metrics. In this setting, Min–Max (without conditional training) achieves better EGT improvements. To simplify the narrative and improve readability, we will reorganize the results to clearly separate RW, Min–Max, and Conditional (without combining them). The freed space will allow us to better explain the intuition behind each behaviour.
>
> **On Figure 4 and the fairness trade-off.**
> We respectfully disagree that this figure is a weakness. The phenomenon highlighted (raising all groups to the worst-performing one) is *not* an artefact of our design but a *mathematical inevitability* of equalized treatment. We believe exhibiting this trade-off is essential for practitioners: EGT fairness has a cost, and understanding that cost is critical for real-world deployment.
>
> **Binary sensitive attribute and intersectionality.**
> We acknowledge this limitation. Our choice was driven by the desire for comparable, cross-modality experiments (FFHQ vs Wikipedia biographies), where intersectional attributes are not aligned. We initially considered age as a second attribute: it is well defined on FFHQ, but not for biographies. Given this is a paper balancing theoretical contributions and experimental validation, we chose a clean setup. We are eager to consider intersectional extensions in future work.
>
> **Error bars and experimental details.**
> We fully agree with the reviewer: error bars should be included, and we will add them in the final version. Likewise, we will move key evaluation details (TopP&R configuration, sampling parameters, oracles, sample sizes) from the appendix into the main text, using the extra space available if accepted. These are essential to communicate the depth of the experiments.
>
> ---

---

> > ### Author Response · Authors · 2025-11-21
> > ****Answers to reviewer questions****
> >
> > **(1) Choice of f-divergence.**
> > The choice of divergence depends on the goals of the downstream application. Some tasks benefit from broad coverage of the data distribution, while others demand high precision.
> >
> > - When *diversity* is important, such as in data augmentation or creative generation for images or text, f-divergences that emphasize coverage are appropriate.
> > - When *high fidelity* is required, for example in medical or scientific settings, divergences that prioritize quality are more suitable.
> > - For practical guidance, Precision and Recall can be expressed as f-divergences and have been effective across both image and text modalities. By linearity of f-divergences, any positive linear combination α·Precision + β·Recall (α,β > 0) is also an f-divergence and allows tuning quality/diversity trade-offs. We therefore recommend them as reasonable starting points, while encouraging users to adjust or combine divergences based on the requirements of their application.
> >
> > **(2) Sensitivity to the oracle.**
> > During development we tested several classifiers (ResNet100, DinoV2, and additional backbones). All achieved high accuracy (≈ 99%, the error generally corresponds to images of babies), suggesting classifier variance is unlikely to drive fairness discrepancies in our experiments.
> >
> > **(3) Multi-attribute / intersectional EGT.**
> > EGT naturally extends to multiple sensitive attributes, since “a’’ can represent tuples. The challenge is *numerical stability*: PR metrics require thousands of samples per class to be reliable, so intersectional subgroups must have enough samples in the reference set. Sample efficiency therefore becomes the limiting factor, not the definition itself.
> >
> > We thank the reviewer again for their constructive feedback and insightful comments, which help us clarify the theory–experiment interface and strengthen the exposition of the paper.

---

### Official Review · Reviewer_ATTV · 2025-11-02

**Soundness:** 3
**Presentation:** 2
**Contribution:** 2
**Rating:** 6
**Confidence:** 4

**Summary:**

The paper focuses on the quality disparity issue of generative models, which is becoming a very crucial area within the field of AI fairness. A main contribution of the paper is to introduce a novel group fairness definition called equalized generative treatment (EGT), which designed to monitor quality gaps among different groups using f-divergence. The paper provides several theoretical insights related to the proposed metrics. In experiments, the paper compares several fairness algorithms in terms of the existing fairness metrics and proposed metric, for both diffusion models and language models.

**Strengths:**

- The paper mainly discusses the quality gap issue of generative models, which becomes a significant area within AI fairness.
- A novel group fairness metric for quality gaps is proposed, grounded in theoretical insights. This metric offers the advantage of adapting various f-divergence-based measures.

**Weaknesses:**

- The paper could contain more clear explanations on the novelty and effective of their "methods".
   - The algorithms used in the experiments to reduce the f-divergence gap appear to be relatively straightforward adaptations of existing fairness algorithms. They do not seem specifically designed for effectively mitigating the f-divergence gap. In addition, based on the current results and discussions, it remains unclear when each method is relatively more effective for addressing the generation quality gap.
   - Conditional training demonstrated pretty strong overall results in mitigating the quality gap, which raises the question of whether this improvement comes from the f-divergence-based insight or from the inherent effectiveness of conditional training itself. Although the paper notes that "conditional training does not automatically improve fairness gaps in terms of EGT", further investigation using state-of-the-art conditional training methods could more definitively clarify their impact on the quality gap in generative tasks.

- While there's some discussion of existing fairness metrics (both classification-based and quality-based), a more comprehensive correlation study between the proposed metric and the existing ones would further clarify the significance of the new metric.


Minor:
- Typo in Introduction: A missing white space between 1st and 2nd sentences.
- The paper might consider adding an explanation of the abbreviation EGT in the introduction section.

**Questions:**

The questions are merged in the above sections.

---

> ### Author Response · Authors · 2025-11-21
>
> We sincerely thank the reviewer for the time and constructive feedback provided. We are grateful for the positive assessment of the motivation behind our work and for the insightful comments that helped us clarify the scope and contributions of the paper.
>
> **Clarifying our contribution.**
> We apologize for the misunderstanding regarding the intention of our experimental section. Our goal is *not* to propose new fairness-enhancing methods. Instead, we aim to study how existing methods—originally designed for proportion-based fairness—behave with respect to our newly introduced criterion, Equalized Generative Treatment (EGT). In the camera-ready version, we will rephrase and expand the relevant sections to make this intent explicit.
>
> **Discussion of the three existing methods we evaluate.**
>
> - **Reweighting (RW).**
>   This approach has *no direct theoretical implications* for EGT, as it was designed for criteria based on group proportions. Our objective was to show empirically that enforcing EGO or MGO during training may fail to reduce disparities in EGT, and in some cases may even worsen them. We will revise the text to emphasize this intended demonstration.
>
> - **Conditional training.**
>   While conditional training does not enforce EGT, it *disentangles* the loss across groups. This avoids the pitfall highlighted theoretically in Theorem 3.1: a model may achieve low global divergence even when the error is concentrated on a single group. Conditional models prevent this collapse by ensuring that each group's conditional distribution is learned separately. Thus, conditional training helps indirectly reduce the risk of extreme EGT disparities. We will expand the corresponding discussion for clarity.
>
> - **Min–Max optimization.**
>   This is the *only* method among the three that explicitly optimizes the worst f-divergence across groups, directly targeting EGT at the training-loss level. As noted by the reviewer, the improvement on EGT is visible but less pronounced than expected. This is due to the fact that we finetune pretrained models rather than train from scratch (which is a more realistic setup given the prohibitive computational cost of full retraining). We will clarify this constraint and better explain the resulting behavior.
>
>
>
> **Correlation between proportion-based metrics and quality-based metrics.**
> The reviewer raises an excellent point. The main purpose of our paper is precisely to show that no meaningful correlation exists between proportion-based fairness (e.g., EGO, MGO) and quality-based fairness (EGT). Theoretically, Proposition 3.1 already establishes that one may obtain *any* value of δ‑EGT while keeping δ‑EGO and δ‑MGO equal to zero. To support this claim empirically, we thank the reviewer for the suggestion and will add the following correlation analysis to the revised version:
>
> |       |   $\delta$-P |   $\delta$-R |   $\delta$-PR |   $\delta$-FID |   $\delta$-MGO |   $\delta$-EGO |
> |-----------|-----------|-----------|------------|-------------|-------------|-------------|
> | $\delta$-P   |     1     |     0.829 |      0.931 |       0.025 |       0.035 |      -0.175 |
> | $\delta$-R   |     0.829 |     1     |      0.981 |       0.168 |      -0.1   |      -0.08  |
> | $\delta$-PR  |     0.931 |     0.981 |      1     |       0.14  |      -0.038 |      -0.14  |
> | $\delta$-FID |     0.025 |     0.168 |      0.14  |       1     |      -0.197 |       0.2   |
> | $\delta$-MGO |     0.035 |    -0.1   |     -0.038 |      -0.197 |       1     |      -1     |
> | $\delta$-EGO |    -0.175 |    -0.08  |     -0.14  |       0.2   |      -1     |       1     |
>
> This correlation table illustrates that EGT behaves fundamentally differently from proportion-based fairness criteria and cannot be inferred from them.
>
> We thank the reviewer again for the helpful comments, which significantly contribute to strengthening the clarity and impact of the paper.

---

### Meta-Review · Area_Chair_brTp · 2025-12-18

**Summary:**

While the reviewers agree the paper proposes a novel group fairness measure for quality gaps among groups using f-divergence, there are also concerns on the triviality of the claims, unfairness mitigation, correlation with other fairness measures, baselines, and experiments. In particular, all the reviewers point out that there is no fairness enhancing method being proposed. While the authors clarify that the contributions of the paper are to only propose the fairness measure and show that no existing method improves it properly, this means the scope and contributions of the paper may be too narrow and that the reviewers were somewhat mislead to believe that there should be a mitigation algorithm. Whether or not the scope is broad enough for ICLR is debatable, and it looks like the reviewers are split where the most negative reviewer is the most confident. If the paper were to be submitted as is, it might be better to target a conference that specializes in Trustworthy AI. Otherwise, it is recommended to also explore how to actually improve the new fairness for completeness.

**Reviewer Concerns:**

The authors addressed most of the reviewer concerns, but the fundamental problem on whether the scope and contributions are appropriate (i.e., is it enough to propose a new fairness measure and show that existing methods don't work, or should the paper also propose a mitigation algorithm) remains.

**Reviewer Scores:**

Overall, there is no indication that the scores would have changed.
* Reviewer ATTV (rating 6) seems satisfied with the rebuttal.
* Reviewer hkXX (rating 6) seems satisfied with the rebuttal.
* Reviewer hajo (rating 2) had a long discussion with the authors, but ultimately did not suggest a score increase.

---

### Decision · Program_Chairs · 2026-01-26

Reject